

# Snow microphysical processes in orographic turbulence revealed by cloud radar and in situ snowfall camera observations

Anton Kötsche[1], Maximilian Maahn[1], Veronika Ettrichrätz[1], and Heike Kalesse-Los[1]

[1]Leipzig Institute for Meteorology (LIM), Leipzig University, Leipzig, Germany

**Correspondence:** anton.koetsche@uni-leipzig.de

**Abstract.** Turbulence influences snow microphysics and precipitation formation, while simultaneously degrading polarimetric radar measurements through broadening of the canting angle distribution. We investigate these interactions in the Colorado Rocky Mountains, where an orographic turbulent layer consistently forms in the lee of Gothic Mountain during precipitation events. To isolate microphysical signals from turbulence-induced artifacts, we apply a novel framework contrasting radar observations above and below the turbulent layer. The dataset combines polarimetric W-band and collocated Ka-band radar measurements with surface in situ observations from the Video In Situ Snowfall Sensor (VISSS). All observations were collected during the CORSIPP project, part of the ARM SAIL campaign (winter 2022/2023).

Aggregation is identified as a dominant process within the turbulent layer, occurring primarily between –12 and –15 °C. It is responsible for reflectivity ($Z_e$) increase of up to $20\,\mathrm{dBZ\,km^{-1}}$ and reduction of the mean particle fall velocity. Enhanced $K_{DP}$ and $sZ_{DR_{max}}$ further suggest secondary ice production through ice-collisional fragmentation, generating anisotropic splinters. Riming also occurs frequently, with $Z_e$ increases up to $15\,\mathrm{dBZ\,km^{-1}}$ and systematically increasing mean particle fall velocity. Riming inside the turbulent layer was often observed at temperatures below -10 °C, indicating the presence of supercooled liquid at cold conditions. Statistical analysis revealed that the turbulent layer is frequently collocated with supercooled liquid water layers near the Gothic Mountain summit.

Our findings demonstrate how radar polarimetry may be safely used to investigate microphysical processes inside a turbulent layer and highlight the impact of orographic turbulence on snow microphysics and precipitation enhancement.

## 1 Introduction

Mountainous regions play a central role in the earths water budget, acting as the Earth's water towers, and are especially vulnerable to changes in the amount of precipitation (e.g Li et al., 2023, and references therein). Understanding precipitation generation and distribution in mountainous regions depends on understanding the microscale dynamics and microphysics that transform condensed water into precipitation.

In the context of precipitation formation in mountainous regions, turbulence has been a key area of research in recent years. Turbulence in mountainous areas frequently occurs due to wind shear, which involves a significant variation in wind speed and direction across a small vertical distance. This may for example occur at the interface between a blocked low level flow inside a valley and the unblocked cross-barrier flow aloft (e.g. Ramelli et al., 2021; Kötsche et al., 2025). Blocking caused by



mountains is a very complex topic and further insight may be found in the work of, for example, Smolarkiewicz and Rotunno (1989) and Smolarkiewicz and Rotunno (1990). Along the flanks of a blocking mountain, a tip jet can form which features enhanced wind speeds and hence wind shear. Downstream of the blocking mountain, a wake, recirculating flow or general deceleration of the wind is found (Petersen et al., 2005). This recirculating or converging flow in the lee of blocking mountains

also leads to pronounced moisture convergence (e.g. Bhushan and Barros, 2007), which consequently may lead to cloud and precipitation formation.

Various studies in recent years focused on the implications of turbulence on precipitation formation. Lee et al. (2014) demonstrated that turbulence boosts supersaturation, leading to faster ice crystal growth through deposition. Additionally, ice particles collide more effectively with supercooled droplets, enabling them to grow into small graupel particles. Consequently,

turbulence leads to a relatively large number of small-sized graupel particles. Aikins et al. (2016) also argued that in cold post frontal continental environments (temperatures below -15 °C), turbulence plays a significant role in promoting snow growth mainly via deposition and aggregation. Houze and Medina (2005), Houze Jr. (2012) and Medina and Houze (2015) discovered that turbulent updraft cells can generate areas of elevated liquid water content (LWC), which promote riming and coalescence while also enhancing aggregation due to variations in the particle fall velocities. Grazioli et al. (2015) presented measurements

gathered in an inner-Alpine valley, they confirmed that if a turbulent layer persists for several hours, sustaining supercooled liquid water (SLW) production, it prolongs riming and allows for large snow accumulations at ground level. Conversely, Aikins et al. (2016) found no SLW near the turbulent cells when conducting measurements in the Sierra Madre Range in Wyoming. They suggested that the absence of liquid water is likely because the high concentration of ice particles from the cloud above rapidly depletes water vapor, preventing liquid drop formation. Ramelli et al. (2021), consistent with previous studies, reported

changes in the microphysical characteristics of clouds in the turbulent shear layer. These modifications included alterations in particle shape or density and increased ice growth. Similar to Grazioli et al. (2015), their measurements were conducted within an inner-Alpine valley. They furthermore speculate that collisions of fragile ice crystals with large rimed ice particles act as a secondary ice production (SIP) mechanism. Billault-Roux et al. (2023) also reported new ice formation associated with updrafts and turbulence within the Swiss Jura Mountains. They concluded that turbulence and updraft favor riming and SIP

through ice–ice collisions of these newly rimed particles. Chellini and Kneifel (2024) analysed a long-term dataset collected at AWIPEV observatory in Ny-Ålesund. They discovered that higher eddy dissipation rate (EDR) as a measure of in-cloud turbulence is associated with increasing size of ice particles at dendritic-growth temperatures and likely also aids increasing SIP via fragmentation of dendritic structures. For temperatures above -10 °C, Chellini and Kneifel (2024) found that turbulence increases riming rates, suggesting that riming in shallow liquid layers is a fundamentally turbulent process. Evidence of SIP

through ice-collisional fragmentation in connection with an orographic turbulent layer was also reported in Kötsche et al. (2025).

While many studies have highlighted riming, aggregation, and secondary ice production (SIP) as key processes in turbulent layers, they often relied on case studies (Grazioli et al., 2015; Aikins et al., 2016; Billault-Roux et al., 2023; Kötsche et al., 2025) or datasets with limited temporal coverage (Medina and Houze, 2015). Furthermore, direct, collocated in situ validation

of hydrometeor type and shape remains rare in remote sensing studies. Finally, turbulence is causing particles to be more





randomly oriented and is hence increasing the width of the canting angle distribution of particles (Klett, 1995; Garrett et al., 2015). Polarimetric radar variables such as differential reflectivity ($Z_{DR}$) and specific differential phase ($K_{DP}$) are depended on the width of the canting angle distribution (Ryzhkov et al., 2002; Hubbert and Bringi, 2003; Ryzhkov and Zrnic, 2019), hence more turbulence means lower magnitudes of $Z_{DR}$ and $K_{DP}$ (e.g., Kötsche et al., 2025). Turbulence can degrade the interpretability of polarimetric signals as well as inhibit the use of radar Doppler spectra based retrieval techniques, effectively "blinding" radars within turbulent layers. This aspect has, in our opinion, not received sufficient attention in previous studies.

In this study, we focus on characterizing an orographic turbulent layer and the microphysical processes therein for a field site in the Colorado Rocky Mountains utilizing long-term measurements of a slanted dual-polarimetric W-band radar (LIMRAD94, model RPG-FMCW-94-DP Küchler et al. (2017)), collocated with the Video In Situ Snowfall Sensor (VISSS, Maahn et al. (2024)) and the single-polarization Ka-band ARM Zenith Radar (KAZR, Widener et al. (2012)). To overcome the "blinding" effect of turbulence on polarimetric Doppler radar observations, we do not analyze data from inside the turbulent layer, but rather compare variables above and below the turbulent layer to then draw conclusions on the microphysical processes inside it. The collocated VISSS data provides valuable insights into the properties of the observed hydrometeor populations including rime mass fraction. The data collection was conducted under the **C**haracterization of **O**rography-influenced **R**iming and **S**econdary **I**ce production and the associated impact on **P**recipitation rates using radar **P**olarimetry and Doppler spectra (COR-SIPP) project, a component of the DFG Priority Programme SPP-PROM (Trömel et al., 2021). CORSIPP was integrated into the Atmospheric Radiation Measurement (ARM) Surface Atmosphere Integrated Field Laboratory (SAIL) campaign (Feldman et al., 2023) for which the AMF2 was deployed during the exceptionally snowy winter of 2022/2023 in the Colorado Rocky Mountains. The measurement site at Gothic, next to Gothic Mountain, is strongly influenced by turbulence and hence prompts us to examine turbulence's effects on precipitation formation. LIMRAD94, VISSS, and KAZR data are further supplemented by other nearby remote sensing and in situ instruments deployed during SAIL.

In Sect. 2, we describe the instruments, data, and methods utilized in this study. The EDR retrieval, turbulent layer detection and SWL layer detection are explained in Sect. 2.4, Sect. 2.5 and 2.6, respectively. The novel methodology developed to derive microphysical processes inside the turbulent layer is presented in Sect. 2.7. The results and discussion Section (Sect. 3) contains a case study (Sect. 3.1), a statistical evaluation of the turbulent layer (Sect. 3.2) and an analysis of fingerprints of microphysical processes in the turbulent layer (Sect. 3.3). The summary and conclusion can be found in Sect. 4.

## 2   Data and Methods

In the following sections, the instruments used for this study and their respective variables are described. We provide an overview of the retrievals for the eddy dissipation rate ($EDR$), the liquid layer using lidar data, and the turbulent layer height (TLH). Additionally, we present the methodology for deriving microphysical processes inside the turbulent layer.





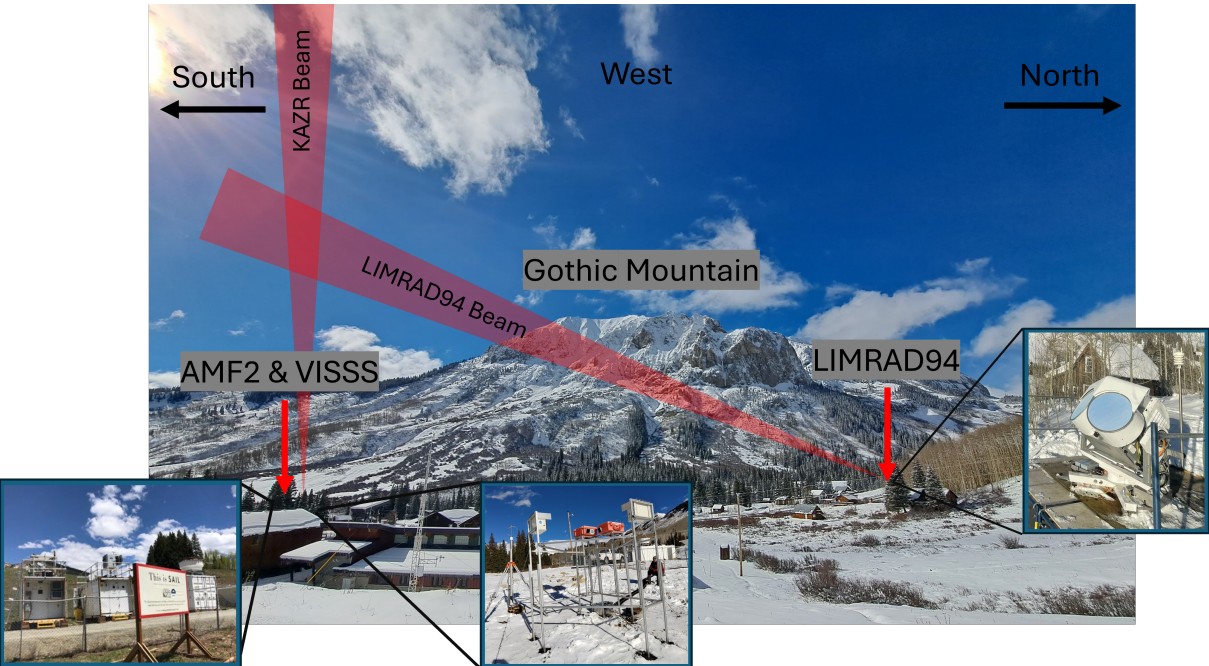

**Figure 1.** The primary measurement site (Gothic, Colorado) during the second winter of the SAIL campaign 2022/23, where the CORSIPP project was conducted.

## 2.1 Radars

### 2.1.1 Overview

Throughout CORSIPP, the dual-polarimetric W-band radar (LIMRAD94, model RPG-FMCW-94-DP Küchler et al. (2017)) was operated within a cold-temperature scanning unit. This installation took place at an elevation of 2905 m at the Rocky Mountain Biological Laboratory (RMBL), approximately 525 m from the second ARM Mobile Facility (AMF2) in Gothic, Colorado. LIMRAD94 was positioned near the RMBL Ore House from November 15, 2022, to June 5, 2023. Observations with LIMRAD94 were made at a fixed elevation of 40° and at an azimuth angle of 151° pointing towards the VISSS and the AMF2 as illustrated in Fig.1. For additional technical specifics regarding the deployment of LIMRAD94 and VISSS during CORSIPP, please refer to Kalesse-Los et al. (2023) and Kötsche et al. (2025). As part of SAIL, the ARM KAZR (Widener et al., 2012) was stationed at the AMF2 at 2889 m ASL. Additionally, the ARM 915 MHz Radar Wind Profiler (RWP915, Muradyan and Coulter (2020)) was installed at the AMF2 in November 2022, providing profiles of horizontal and vertical wind with a resolution of 1 hour.



### 2.1.2 Polarimetric radar data

LIMRAD94 has the capability to measure spectral polarimetric variables such as differential phase shift and differential reflectivity, among others. The polarimetric calibration for LIMRAD94 was conducted at the onset of the field campaign, coinciding with a period of vertically oriented measurements as outlined in Myagkov et al. (2016).

The differential phase shift $\Phi_{DP}$ (°) consists of a backscatter and a propagational part, where the propagational part is called specific differential phase $K_{DP}$ (° km$^{-1}$). $K_{DP}$ depends on the shape, orientation, concentration, density and the size of hydrometeors. Elevated concentrations of dense ice particles that exhibit greater anisotropy in shape and possess uniform orientation result in increased values of $K_{DP}$. It was furthermore shown by Kötsche et al. (2025) that up to 20 % of W-band $K_{DP}$ can be attributed to large aggregates. Additionally, $K_{DP}$ is affected by turbulence due to its reliance on the width of the canting angle distribution ($\sigma$) as shown by Ryzhkov et al. (2002); Ryzhkov and Zrnic (2019). As turbulence intensifies, the magnitude of $\sigma$ increases, consequently leading to a reduction in $K_{DP}$.

The differential reflectivity $Z_{DR}$ (dB) is commonly defined as the ratio between the radar reflectivity measured at horizontal polarization and at vertical polarization. At W-band, the largest contribution to elevated $Z_{DR}$ values in the ice phase typically comes from dense, anisotropic ice crystals that are small enough to be within the Rayleigh scattering regime, and often found on the slower side of the radar Doppler spectrum. Unlike $K_{DP}$, $Z_{DR}$ is unaffected by the particle concentration. More spherical and less dense particles result in a reduced $Z_{DR}$. Since $Z_{DR}$ is reflectivity-weighted, its magnitude is often reduced by larger, spherical particles (e.g., Oue et al., 2018). To avoid this masking issue when using $Z_{DR}$, we also determine the maximum spectral $Z_{DR}$ value ($sZ_{DR_{max}}$) for each Doppler spectrum. Similar to $K_{DP}$, overall $Z_{DR}$ values are reduced by turbulence through a more random orientation of particles and broadening of the Doppler spectrum. More detailed descriptions of these polarimetric variables and the influence of turbulence on them can be found in Kötsche et al. (2025).

### 2.2 VISSS

The first generation VISSS was installed next to the AMF2 facility, in the line of sight of LIMRAD94 with a horizontal distance to LIMRAD94 of approximately 550 m (Fig. 1). VISSS1 has a pixel resolution of 58.832 µm px$^{-1}$, a frame rate of 140 Hz, and an observation volume of 75.2 x 75.2 x 60.1 mm$^3$ (Maahn et al., 2024). This yields an observational volume of 0.000339 m$^3$. VISSS level2match variables (see Fig. 2 in Maahn et al. (2024)) used in this study include particle size distributions (PSD), $D_{32}$, complexity, and total number concentration ($N_{tot}$). $D_{32}$ is the ratio of the third to the second measured PSD moment. Assuming that particle mass is proportional to the particle maximum dimension squared, $D_{32}$ is a proxy for the mass-weighted mean diameter of the particle population (Maahn et al., 2015). Following Schmitt and Heymsfield (2014), the complexity $c$ is derived from the ratio of the particle perimeter $p$ to the perimeter of a circle with the same area $A$ :

$$c = \frac{p}{2\sqrt{\pi A}}. \tag{1}$$



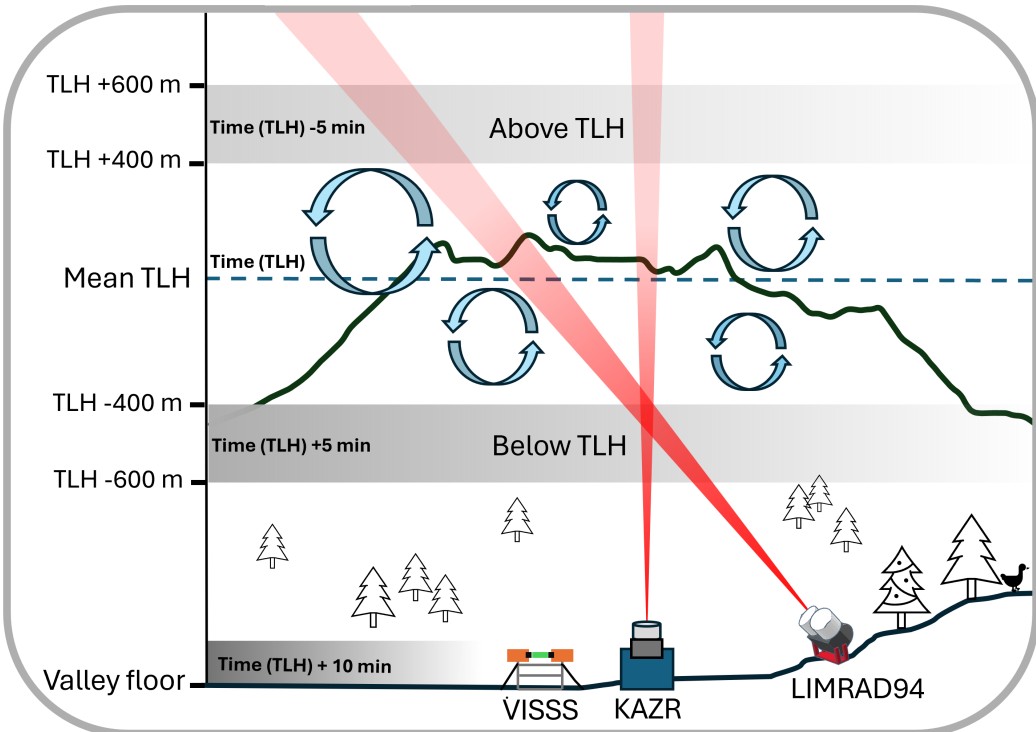

**Figure 2.** Methodology to analyze microphysical processes inside the turbulent layer described in Sect 2.7. Shown is a schematic of the measurement site including LIMRAD94, KAZR, VISSS and their measurement volumes. Blue arrows indicate turbulence which most frequently occurs around the summit height of Gothic Mountain (black line in the background). The dashed blue line indicates the mean turbulent layer height. Grey areas above TLH and below TLH mark the area in which radar data are averaged to avoid contamination of the measurement volume by turbulence.

## 2.3 Additional data products

Vertical temperature profiles on site are provided by the ARM interpolated sonde and gridded sonde VAP (Fairless et al., 2021).
The sounding data, collected on site through two radiosonde launches per day (11 and 23 UTC), is transformed into continuous daily files with 1-minute time resolution and combined with ARM 3-channel microwave radiometer temperature data. Data from the high resolution spectral lidar (HSRL) installed at AMF2 (Goldsmith, 2016) was used to detect SLW. Particle number concentrations for the whole SAIL campaign period were measured by the ARM Laser Distrometer (Bartholomew, 2020), an OTT Parsivel2 (from now on referred to as Parsivel2). The cloud base height was retrieved by the ARM ceilometer ((Morris,
2016)). Rime mass fraction $M$ was calculated using VISSS particle size distributions (PSD) and LIMRAD94 $Z_e$ with the approach of Maherndl et al. (2024, 2025).



## 2.4 Turbulence eddy dissipation rate retrieval ($EDR$)

As a proxy for atmospheric turbulence, the eddy dissipation rate $EDR$ quantifies the rate at which turbulent kinetic energy is lost to viscous forces (Foken and Mauder, 2024). Elevated $EDR$ values are associated with more intense turbulence. In this study, $EDR$ was estimated from mean Doppler velocity (MDV) measurements obtained by KAZR, aggregated over five-minute intervals following the method outlined in Vogl et al. (2024). To isolate regions of significant turbulence, we applied an empirical threshold of $10^{-3}\,\mathrm{m^2\,s^{-3}}$, consistent with the criterion used by Vogl et al. (2022). It is important to note that during the CORSIPP campaign, the radar beams of KAZR and LIMRAD94 were not perfectly aligned (Fig. 2). As a result, the volumes from which KAZR-$EDR$ and LIMRAD94 measurements were collected are horizontally offset, with the separation increasing with altitude (580 m horizontal distance at Gothic Mountain summit height).

## 2.5 Turbulent layer detection

The turbulent layer height (TLH) was detected using $EDR$ (described in 2.4). Specifically, we use the vertical profile of $EDR$ and calculate the mean $EDR$ weighted TLH as follows:

$$TLH = \frac{\sum_i h_i \cdot EDR_i}{\sum_i EDR_i} \tag{2}$$

where $h$ is the height. As we are interested in the turbulence directly induced by orography, we only include $EDR$ data from up to 2800 m AGL for our calculations. Because $EDR$ requires $MDV$ measurements from KAZR, TLH is detectable with our approach only if KAZR can measure sufficient tracers like cloud droplets, ice particles or precipitation.

## 2.6 Liquid layer detection

A liquid layer was detected when the HRSL attenuated backscatter coefficient at 1064 nm exceeded > 1e-5 $\mathrm{m^{-1}\,sr^{-1}}$ and the linear depolarization ratio was < 0.05, as the depolarization of water droplets should be close to 0 according to (Vaisala, 2021). The lidar data was filtered using a signal to noise ratio threshold of 10 dB. In the following, we use LLB to refer to the liquid-containing layer base height. We acknowledge that the LLB will generally be consistent with the ceilometer-derived cloud base height, since both instruments detect liquid layers as proxies for cloud base. Nevertheless, retrieving the LLB directly from HSRL adds value: it provides an independent product for cross-validation with the ceilometer, and it allows cloud-base detection to be performed using only the HSRL data set. This ensures consistency for studies that rely exclusively on HSRL observations.

## 2.7 Deriving microphysical processes inside the turbulent layer

To analyze microphysical processes within the turbulent layer, we are comparing KAZR and LIMRAD94 data above and below the turbulent layer rather than using data directly influenced by turbulence. Data between Dec 1, 2022 and Feb 7, 2023 was used for the analyses because of the continuously available constant elevation measurements of LIMRAD94 and frequent precipitation cases during that time period. A schematic of the methodology is shown in Fig. 2.



The radar data of KAZR and LIMRAD94 is first filtered using a signal to noise ratio threshold of $10\,\mathrm{dB}$ and resampled into 5-minute mean intervals, matching the temporal resolution of $EDR$. Data from range gates where the nearest $EDR$ measurement exceeded the turbulence threshold of $10^{-3}\,\mathrm{m^2\,s^{-3}}$ are excluded from the analysis. Subsequently, range gates located between 400 and 600 m both above and below the TLH are identified, and the mean value of the data within these specific range gates is computed. This calculation is conducted solely when the TLH attains a minimum of 600 m above ground level (AGL). The distances from the turbulent layer are based on empirical observations of the vertical extent of the turbulent layer. Afterwards, for each timestep where a turbulent layer is identified, the method computes the difference between radar observations below the TLH at the timestep T(TLH) +5 min and those above the TLH from 5 minutes prior at T(TLH) -5 min. The empirical time lag of 5 minutes (per 500 m vertical distance) is introduced to compensate for the fall velocity of hydrometeors. To complement the radar data, collocated VISSS data are selected at T(TLH) +10 min to provide additional context on the microphysical properties of the PSD. A time lag of T(TLH) +10 min is chosen because we have to take into account the time in which the particles sediment from the radar range gates below the turbulent layer to the surface. The result of this analysis is shown and discussed in Sect. 3.3.

We acknowledge that this method relies on the assumption of horizontal and temporal homogeneity of precipitation, particularly important in the case of slanted radar measurements. In our case study, the prevailing wind direction is westerly, while LIMRAD94 is oriented toward the southeast. As a result, it is clear that we do not observe the exact same hydrometeors throughout their descent. Additionally, the horizontal distance between corresponding range gates of LIMRAD94 and KAZR increases with altitude. At summit height of Gothic Mountain, the horizontal distance between the range gate centers of LIMRAD94 and KAZR is about 580 m. The instruments also have different half-power beam widths, leading to different measurement volumes. These factors further necessitate the assumption of horizontal homogeneity and the use of temporal averaging. However, near the surface and below the turbulent layer, the measurement volumes of both radars are more closely aligned. The vertical distance from the measured volume below TLH to VISSS is approximately 500 m (assuming a mean TLH of around 1000 m AGL). Since the most significant changes in precipitation, along with potential inhomogeneities, are expected to occur below the turbulent layer, we argue that the combined dataset remains highly informative and suitable for the intended analysis.

## 3 Results and Discussion

In the following Section, we present a case study featuring a distinct turbulent layer. Additionally, we include a statistical analysis of the turbulent layer, as well as CBH and LLB height during the whole time frame of the SAIL campaign. Furthermore, we discuss the fingerprints of microphysical processes in the turbulent layer using KAZR and LIMRAD94.

### 3.1 Case Study on February 21, 2023

The case study on February 21, 2023 provides an example on how the turbulent layer can locally alter the temperature and moisture profile in the atmospheric boundary layer, create a liquid layer and produce precipitation. On this day, the site is



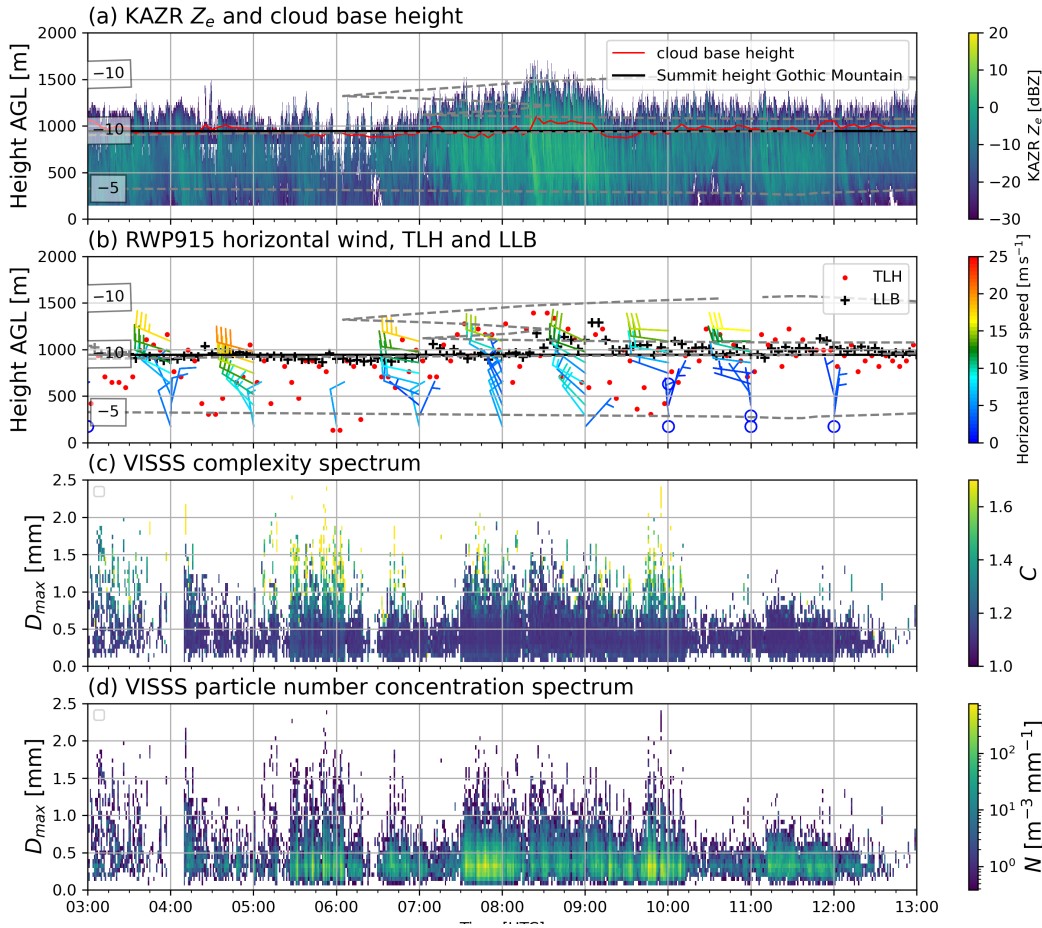

**Figure 3.** Case study on Feb. 21 2023. (a) KAZR $Z_e$ and ceilometer cloud base height (solid red line) with the summit height of Gothic Mountain (black) overlaid. Grey dashed lines are the isotherms with respective labels on the left. (b) RWP915 horizontal wind barbs (color shading) with turbulent layer height (TLH, red dots) and liquid layer base height (LLB, black crosses). (c) VISSS complexity spectrum as a function of maximum particle diameter ($D_{max}$). (d) VISSS particle number concentration spectrum as a function of $D_{max}$.

under slight high pressure influence, located between a high pressure system to the south-west and a low pressure system further east in the Denver front range area. This pressure gradient causes a west to north-westerly cross-barrier flow with up to 10-20 m s$^{-1}$. An overview plot for the case is shown in Fig. 3. Panel (a) shows the KAZR reflectivity and the CBH. During the whole case study, the CBH is located at the approximate summit height of Gothic Mountain. The TLH (panel b) differs in height, which is closely tied to the wind direction. In the first half of the case study until 9 UTC, the wind direction is north-northwesterly, which causes the turbulent layer to form along the northern edges of Gothic Mountain (discussed more thoroughly in Sect. 3.2). The height therefore varies between 250 m AGL and Gothic Mountains summit height. Between 7 and 9 UTC, a stronger shower alters the wind profile, causing TLH of up to 1400 m. After 9 UTC, the westerly wind is blocked



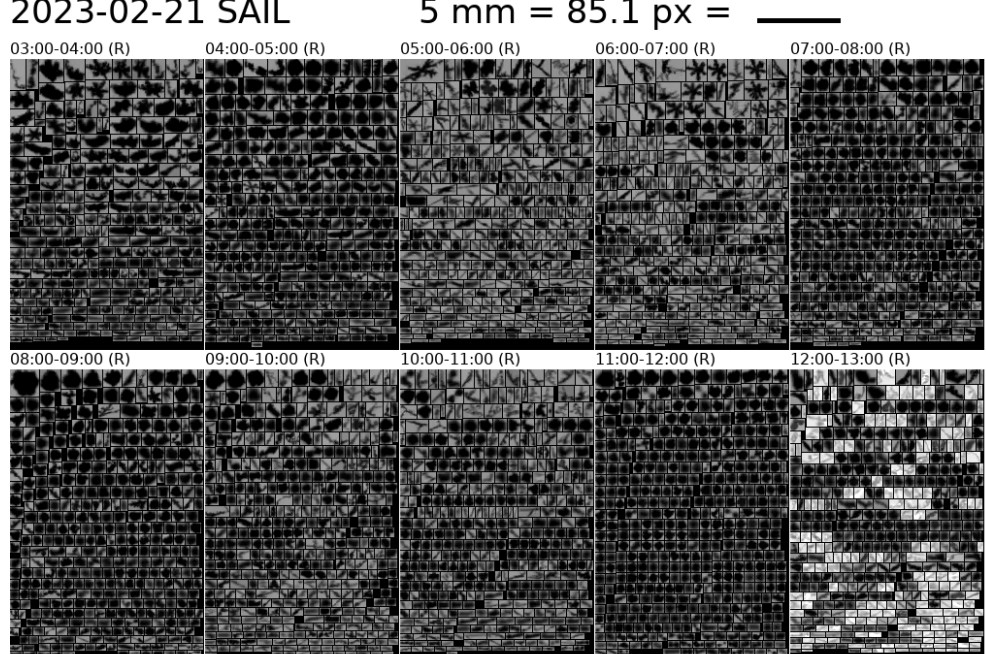

**Figure 4.** VISSS particle images recorded during the case study on February 21, 2023.

by Gothic Mountain, the wind shear along the interface to the unblocked cross-barrier flow causes the TLH to be collocated with the Gothic Mountain summit height. The LLB height is closely tied to the Gothic Mountain summit height for the whole period, generally at the upper edge of the turbulent layer. The liquid layer is likely produced by a combination of moisture
convergence in the lee of Gothic Mountain and constant turbulence producing SLW within updrafts.

The clouds are rather shallow with cloud depths of 250-500 m. However, they are located at a temperature of around -10 °C. This temperature range, favorable for depositional growth of ice particles, combined with vertical air motion induced by turbulence causes these clouds to almost constantly produce precipitation, as shown by the VISSS particle number spectrogram in panel (d). The majority of particles has $D_{max} < 1\,\mathrm{mm}$ and a complexity below 1.2, which indicates graupel particles (panel
(c)). Occasionally, we also see larger particles with complexity exceeding 1.6, which are needles, needle aggregates, and a few dendrites as revealed by VISSS images (see Fig. 4). Especially after 9 UTC, we see a decrease in $Z_e$ towards the ground, indicating ongoing sublimation of precipitation below the turbulent layer. A radiosonde launch during the case study at 11:30 UTC (Fig. 5) reveals how the turbulent layer is modifying the atmospheric boundary layer. There is a radiation inversion at the surface given that 11:30 UTC corresponds to the early morning hours (04:30 MST). Between 500 m and just below
summit height of Gothic Mountain, the boundary layer is well mixed with a dry adiabatic temperature gradient. As solar radiation is not the cause, wind shear is likely mechanically mixing the boundary layer. At summit height of Gothic Mountain, we find a thin layer of enhanced relative humidity, collocated with the average TLH during the time of the radiosonde launch. In this shallow mixed-phase cloud layer, the precipitation detected by VISSS at the ground is forming. This layer is likely



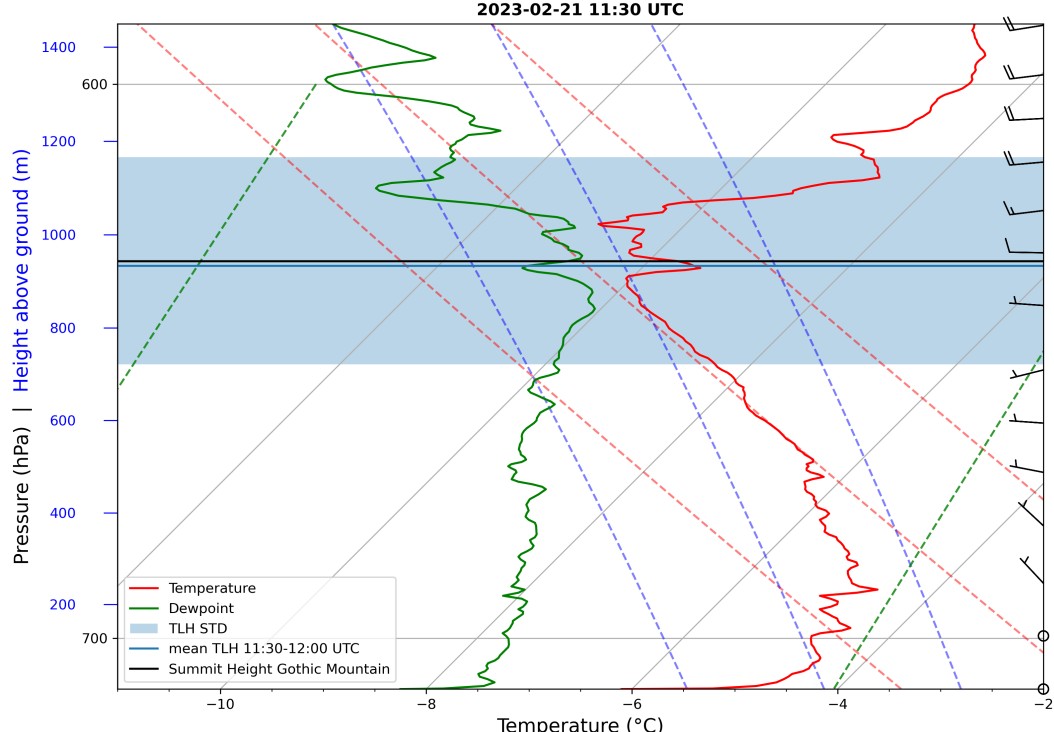

**Figure 5.** Skew-T diagram of Temperature (red line), relative humidity with respect to water (green line) and wind direction and speed from a radiosonde launched during the case study on Feb. 21 2023 at 11:30 UTC. Blue line and blue shaded area mark the mean turbulence layer height and its standard deviation, respectively. The black line marks the summit height of Gothic Mountain.

caused due to a combination of enhanced moisture convergence behind Gothic Mountain due to converging flow (e.g. Bhushan and Barros, 2007), turbulent mixing of the boundary layer with the highest relative humidity at the top of the mixed layer, and constant production of SLW through turbulence. The low relative humidity below this layer enables the sublimation of precipitation, visible in a decrease of KAZR $Z_e$ towards the ground.

## 3.2 Characterization, occurrence and position of the turbulent layer

Although case studies provide significant insights into the microphysical processes associated with the turbulent layer, our objective is to comprehend its impact from a more statistical perspective.

For the whole duration of the SAIL campaign, KAZR and HRSL lidar data from Gothic and hence EDR, CBH, LLB height and TLH are available. In Fig. 6, a statistic of TLH, LLB and CBH between Sep 2021 and May 2023 is shown. In panel (a), the height difference between LLB,TLH and CBH,TLH is plotted (if a turbulent layer was detected). The difference is mostly less then 1000 m, showing a close collocation of LLB, CBH and TLH. Panel (b) illustrates the frequency of CBH, LLB, and TLH during the same period, yet considered separately from one another (LLB and CBH also included if no TLH was present). TLH

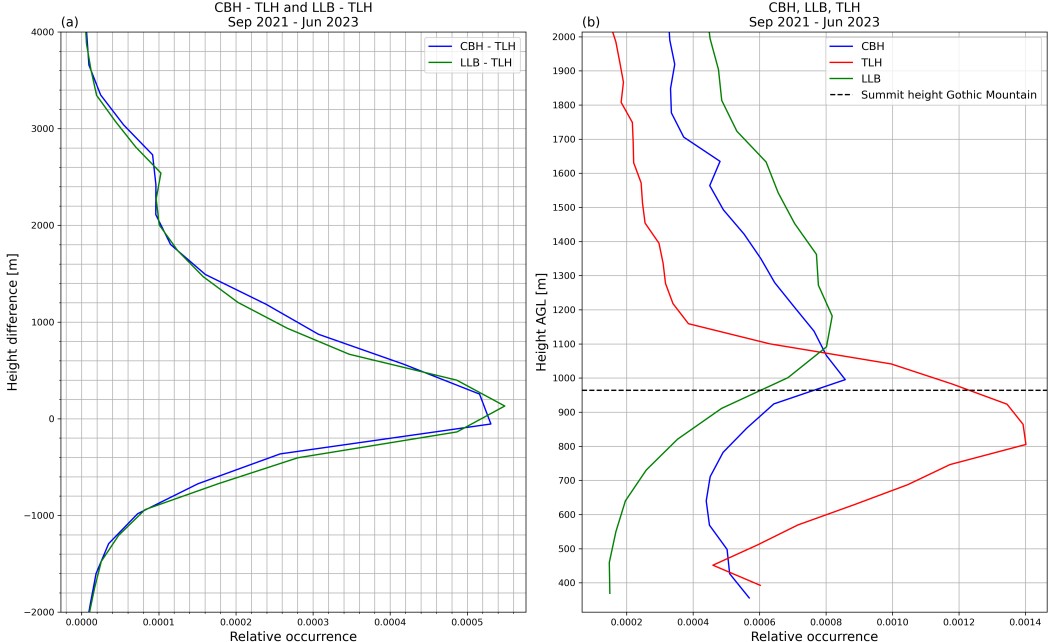

**Figure 6.** Panel a): Statistic of difference between CBH and TLH (blue line) and the difference between LLB and TLH (green line) for the duration of the SAIL campaign from September 2021 to end of May 2023. Panel b): Statistic of turbulent layer height detected as described in Sect. 2.5 (red line), ceilometer cloud base (blue line), liquid layer base height (green line) for the duration of the SAIL campaign from September 2021 to end of May 2023. The black dashed line marks the Gothic Mountain summit height.

occurs most frequently just below the summit height of Gothic Mountain at around 800 - 900 m AGL. This and the collocation of LLB and CBH shown in panel (a) suggests that the turbulent layer plays a major role in cloud formation, possibly through enhanced moisture convergence in the lee of Gothic Mountain. It also implies that the turbulent layer aids the formation of a SLW layer as it was already shown in (Houze Jr., 2012; Medina and Houze, 2015; Ramelli et al., 2021). The LLB/CBH were

245 often occuring a few hundred meters above TLH, this may arise from the fact that we look at the base of the liquid-containing layer vs the mean TLH while the turbulent layer has a vertical extent of a few hundred meters. Liquid water drops, due to their low mass, are most likely found at the top of the turbulent layer as their are being lofted by the turbulent updrafts (Zhu et al., 2023). The position of TLH, CBH and LLB in the shorter time period in which we analyse LIMRAD94 data from Dec 2022 to Feb 2023 did not differ greatly.

In Fig. 7 the HRSL LLB height is plotted against the temperature at LLB height. The figure shows that cloud liquid is frequently found at heights between 800 and 1600 m AGL with the maximum frequency of occurrence mostly in the vicinity of Gothic Mountain Summit and temperatures between -20 and -5 °C, theoretically allowing for riming even at temperatures down to -20 °C.



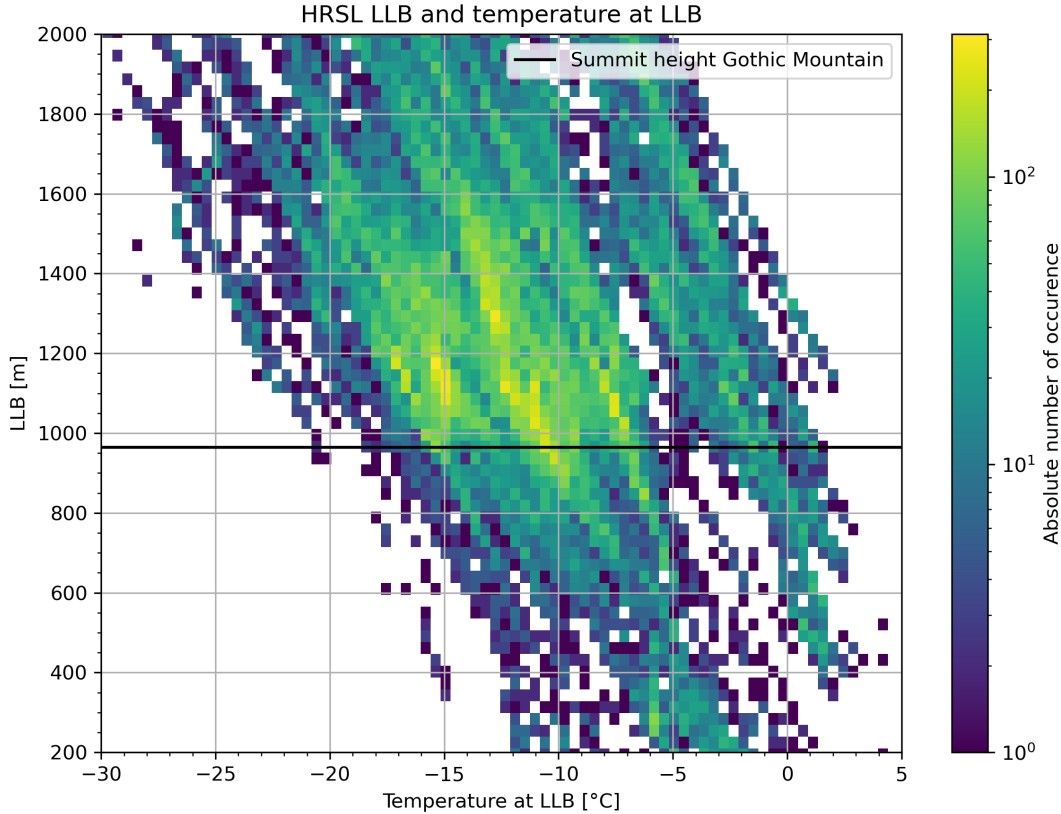

**Figure 7.** 2D histogram of LLB height for the duration of the SAIL campaign from September 2021 to end of May 2023 derived from HRSL measurements, and the respective temperature at LLB height. The black solid line marks the Gothic Mountain summit height.

Due to the availability of the wind profiler RWP915 between Dec 2022 and May 2023, we can relate TLH to the wind
direction in this period. It is important to notice that RWP915 data is only available during precipitation when hydrometeors
serve as tracers for the wind direction. Wind during this time was strongly dominated by south, west and north wind, especially
during significant precipitation events. The position of the measurement site in the East River valley introduces a complex ter-
rain, strongly varying with azimuth direction. To link the occurrence of turbulence to the surrounding orography we combined
the TLH with the wind direction at TLH retrieved by RWP915 and the max. terrain height in a $3\,\mathrm{km}$ perimeter around the
measurement site (see Fig. 8). The highest mountain close to the measurement site is Gothic Mountain, visible as a peak in
orographic height between an azimuth of 240 - 310°. The occurrence of TLH closely resembles the shape of Gothic Mountain.
At wind directions of 240-300°, the turbulent layer is located at summit height of Gothic Mountain or a few hundred meters
above. Slight changes in the wind direction to more southerly or northerly directions causes the TLH to decrease and the tur-
bulence seems to be created along the flanks of Gothic Mountain. During southerly wind directions, a turbulent layer is created
by Snodgrass Mountain which is a second, smaller summit south of the measurement site at an azimuth of 150-200°. During



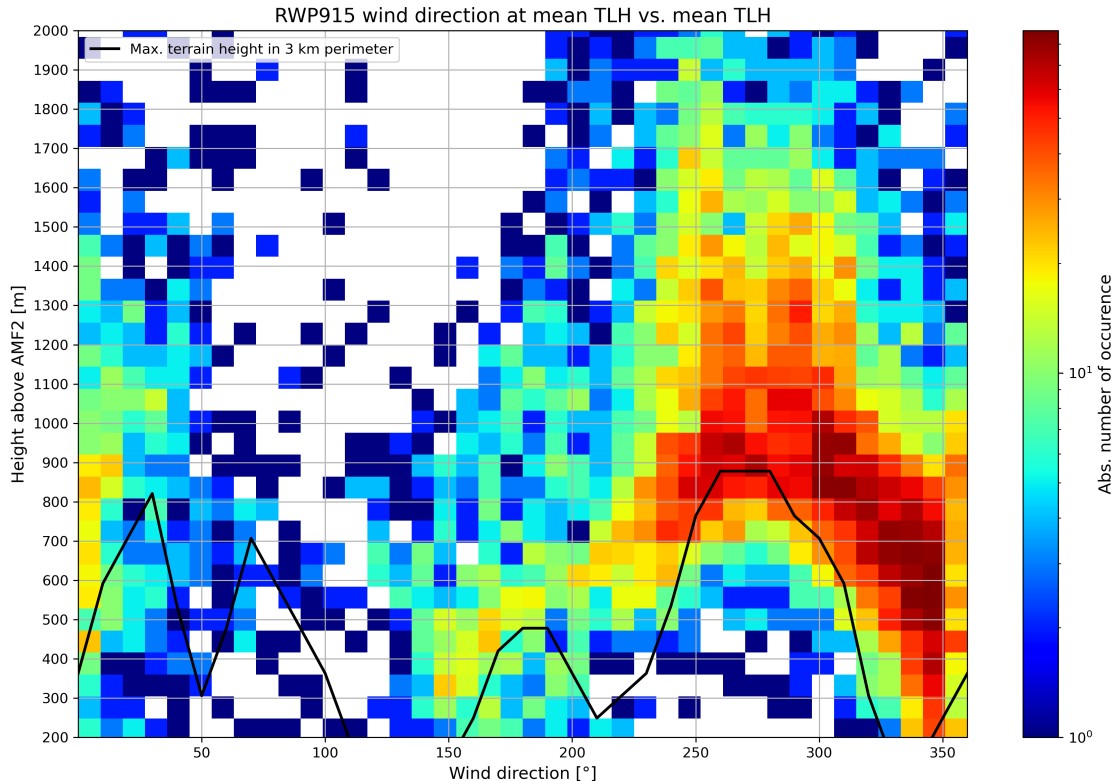

**Figure 8.** Absolute number of occurrence of TLH (retrieved as described in Sect. 2.5) plotted as a function of the wind direction at TLH (measured by the RWP915 between Dec 2022 and May 2023). The black line is the max. terrain height in a 3 km perimeter around AMF2 at the respective azimuth direction.

periods with northerly wind directions, the wind blows down the East River valley rather undisturbed and the turbulent layer appears anywhere between almost surface level and 1000 m above ground. Either the turbulence is formed along the flanks of Gothic Mountain and Avery Peak to the NE (0-50 ° azimuth), or the flow is still influenced by Mt. Bellview, a summit located 7 km to the NNW of Gothic. A turbulent layer is only rarely detected during wind directions between 50 and 150 °, simply 270 because these wind directions are uncommon during precipitation events in that region.

Figure 9 shows temperature at TLH plotted against the total number of hydrometeors ($N_{tot}$) observed by the Parsivel2 at the surface for different synoptic configurations. Specifically, deep and shallow precipitating systems that either do or do not contain a turbulence layer are contrasted. If no turbulent layer was present, the temperature at Gothic Mountain summit height was used. Almost no precipitation was detected without a turbulent layer (panels (c) and (d)), highlighting the crucial role of 275 turbulence in orographic precipitation, as it often accompanies precipitation, even if not directly causing it. The majority of precipitation events stem from deeper precipitation systems and a turbulent layer somewhere between -5 and -15 °C (panel a). The highest $N_{tot}$ is also found in this temperature region. At temperatures between -10 and -20 °C, the turbulent layer is in an espe-




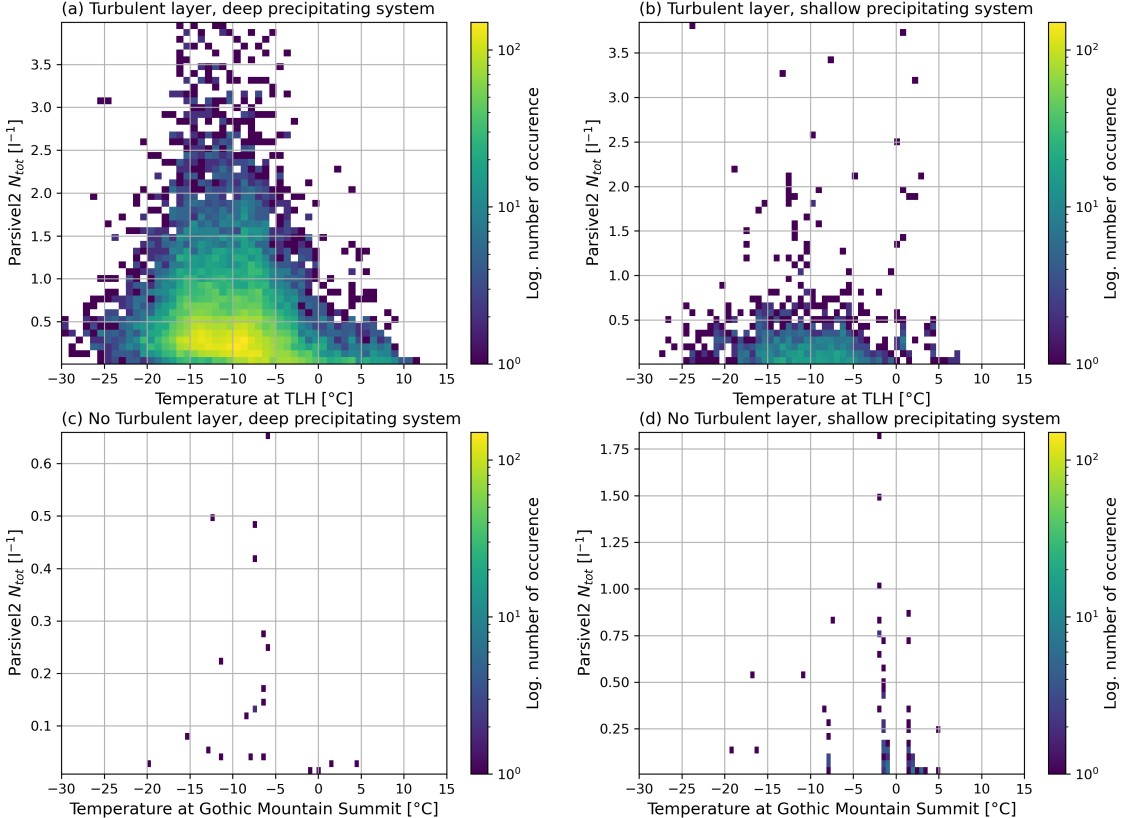

**Figure 9.** Temperature at TLH plotted as function of the Parsivel2 $N_{tot}$ for different synoptic configurations. Data from September 2021 to June 2023 was analyzed. a): A turbulent layer is present, KAZR $Z_e$ between 500 m and 1.5 km above Gothic Mountain summit height exceeds -10 dBZ. b): A turbulent layer is present, KAZR $Z_e$ between 500 m and 1.5 km above Gothic Mountain summit height does not exceed -10 dBZ. c): No turbulent layer is present, KAZR $Z_e$ between 500 m and 1.5 km above Gothic Mountain summit height exceeds -10 dBZ. d): No turbulent layer is present, KAZR $Z_e$ between 500 m and 1.5 km above Gothic Mountain summit height does not exceed -10 dBZ.

cially favorable temperature regime for precipitation formation via the Wegener-Bergeron-Findeisen Process (WBF,(Wegener, 1911; Bergeron, 1935), as liquid water formed through updrafts can quickly transition into the ice phase. The presence of SLW

in this temperature region (see Fig. 7) may further enhance depositional growth and also riming, and SIP processes enhanced by turbulence may aid the formation of new particles, which we will discuss in detail in Sect. 3.3. Most interestingly, precipitation is also detected in the presence of a turbulent layer when only a shallow precipitating system is present. One of these situations was presented in the case study above (Sect. 3.1). Such precipitation is most common at a temperature at TLH between -5 and -20 °C, but features much lower $N_{tot}$ than deep precipitating systems. Precipitation originating from these shallow clouds

may be entirely induced or at least augmented by orographic turbulence; however, definitive evidence remains elusive. We





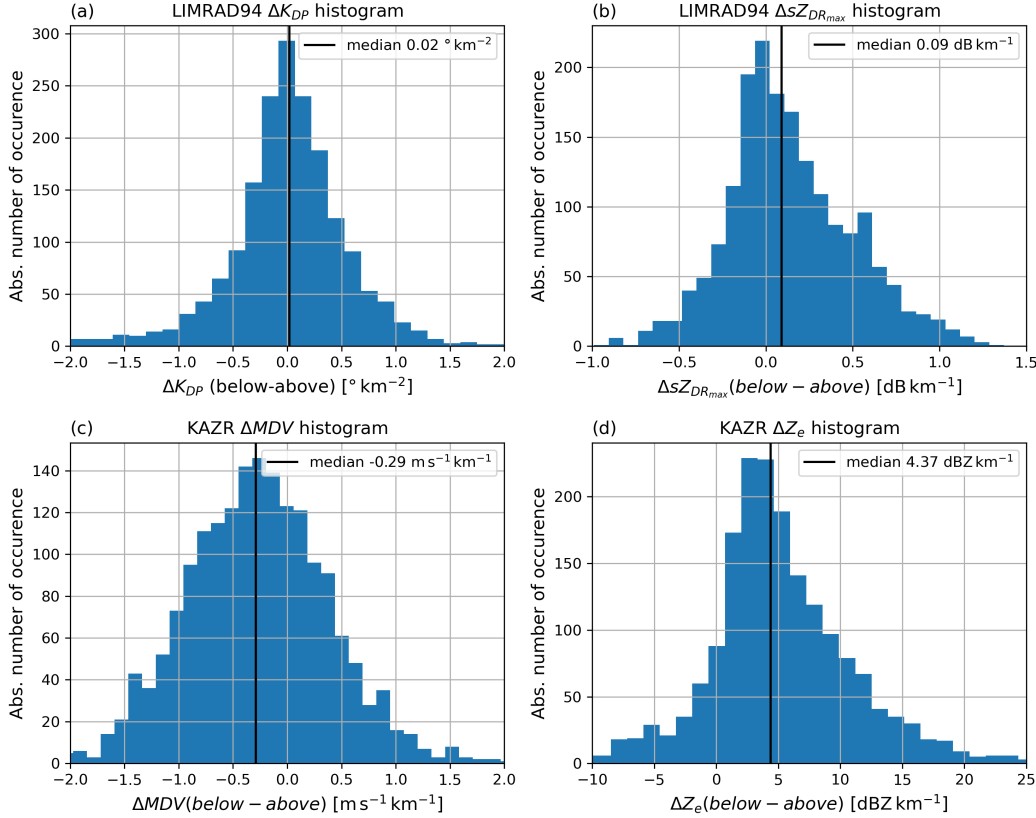

**Figure 10.** Histograms showing the difference of radar variables above-below the turbulent. (a) LIMRAD94 $\Delta K_{DP}$, (b) $\Delta sZ_{DR_{max}}$, (c) $\Delta MDV$, (d) KAZR $\Delta Z_e$.

can, however, provide some eye witness reports, as during the installation of our instruments in Gothic, we had the chance to observe shallow precipitating clouds forming behind Gothic Mountain (see Fig. A1 in the Appendix).

### 3.3 Fingerprints of microphysical processes in the turbulent layer

In this section, we focus on the microphysical processes inside the turbulent layer and the fingerprints they leave in polarimetric
cloud radar data. As mentioned, the problem is that radar observations inside the turbulent layer are masked by the turbulence itself (see Sect. 2.1.2). So instead of looking at radar data inside the turbulent layer, we compare radar variables above and below the turbulent layer and therefrom draw conclusions on microphysical processes inside the turbulent layer (Sect. 2.7). We specifically analyze four variables: $Z_e$, $MDV$, $K_{DP}$ and $sZ_{DR_{max}}$, in a period LIMRAD94 was measuring at constant elevation of $40\,^\circ$ from Dec 2022 to Feb 2023. $MDV$ is defined negative upward.
The histograms of the absolute change from above to below the turbulent layer are shown in Fig. 10. Negative/positive values indicate an decrease/increase of the respective variable within the turbulent layer, respectively. The change in $K_{DP}$ (panel a)



**Figure 11.** 2D histograms showing the difference of radar variables below - above the turbulent layer, the color of each square represents the mean of a third variable. Each square contains at least 4 datapoints. $MDV$ is defined negative upward. All panels divide the x-axis and y-axis into quadrants (I–IV). (a) KAZR $\Delta Z_e$ vs. $\Delta MDV$, color: VISSS $M$, (b) KAZR $\Delta Z_e$ vs. $\Delta MDV$, color: VISSS $D_{32}$, (c) KAZR $\Delta Z_e$ vs. $\Delta MDV$, color: $\Delta sZ_{DR_{max}}$, (d) KAZR $\Delta Z_e$ vs. $\Delta MDV$, color: VISSS $C$, (e) KAZR $\Delta Z_e$ vs. $\Delta MDV$, color: LIMRAD94 $\Delta K_{DP}$, (f) KAZR $\Delta Z_e$ vs. $\Delta MDV$, color: VISSS $N_{tot}$

follows a near perfect Gaussian distribution centered around $0\,^\circ\,\mathrm{km}^{-1}$. $sZ_{DR_{max}}$ (panel b) reveals a tendency towards an increase below the turbulent layer, which gives a first hint on possible SIP mechanisms as $sZ_{DR_{max}}$ increases when non-spherical ice particles are present. Below a turbulent layer, these ice particles might be splinters produced during ice collisional fragmentation. $MDV$ (panel c) also shows a tendency to decrease below the turbulent layer, which means the particles velocity






decreases inside the turbulent layer. This might be connected to the formation of a second, slower falling particle mode within the turbulent layer consisting of splinters, while the cases with increasing $MDV$ point to the formation of rimed particles. $Z_e$ (panel d) mostly increases inside the turbulent layer, which might be explained by increasing particle diameters inside the turbulent layer or also increased density (e.g. caused by riming) or increased number concentration of particles. It is apparent

that we need to exploit the full polarimetric capacity of LIMRAD94 in combination with in-situ measurements by the VISSS to fully understand the microphysical processes inside the turbulent layer.

For that purpose, Fig. 11 illustrates a blend of various radar and VISSS variables, where radar variables are again represented as the difference between measurements above and below the turbulent layer. Surface-based VISSS variables assist in the classification of particle population characteristics. The plots are designed as 2D histograms between KAZR $\Delta MDV$ and

$\Delta Z_e$, while the color of each square represents the mean of a third variable that supplements the plot. Each panel domain is divided into four quadrants (Q) labeled **I** to **IV**. The absolute number of data points in each statistic is shown in panel (i).

**Quadrant I**

Quadrant **I** within each sub-panel comprises data points characterized by an increase in $Z_e$ within the turbulent layer combined with an decrease in $MDV$ (slower particle descent below the TLH compared to above it). This combination is the most

common during the analyzed period as evidenced in panel (i), where the total number of data points is shown.

In panel (a), KAZR $\Delta MDV (below - above)$ is plotted against KAZR $\Delta Z_e (below - above)$ with the retrieved rime mass fraction $M$ as color code. Quadrant **I** shows that an increase of $Z_e$ in combination with an decrease of $MDV$ (which means particles fall slower) below the turbulent layer can be attributed to mostly unrimed particles. An decrease in $MDV$ which is accompanied by a $Z_e$ increase of less than $5\,\mathrm{dBZ\,km^{-1}}$ can generally be found in PSDs containing slightly rimed particles

(light blue colors). In panel (b) **I**, the same radar variables are displayed together with VISSS $D_{32}$ as color. An increase in $Z_e$ in combination with an decrease of $MDV$ is mostly linked to a $D_{32}$ of $3\,\mathrm{mm}$ or higher. These situations are mostly found when also the mean complexity (panel (e) **I**) is higher than 1.7. This points to the existence of aggregates, meaning the $Z_e$ increase in these cases is caused by the increase in the particle diameter ($Z_e \sim D^4$, Moisseev et al. (2017)) through aggregation in the turbulent layer. Towards the lower $Z_e$ increase on the left of quadrant **I**, mean $D_{32}$ is mostly around $2.5\,\mathrm{mm}$ and also the

complexity is slightly below 1.7. These ice particles appear to be slightly more rimed smaller aggregates that do not undergo major size increase inside the turbulent layer, which may indicate they are falling from aloft during deep precipitating systems. This is supported by the higher LWP visbile in panel (c) on the left side of quadrant **I**. Temperatures towards the left of quadrant I are slightly higher (panel f), around or higher than -10 °C, indicating that this might be connected to deeper warm frontal precipitation.

Mean LWP in panel (c) **I** ranges from under $50\,\mathrm{g\,m^{-2}}$ in the outer edges to about $100\,\mathrm{g\,m^{-2}}$ more towards the center and left of the plot. At the same time, the mean complexity (panel (e) **I**) shows an inversely proportional behavior. This relation can be explained as follows: In deep frontal systems, especially along warm fronts, LWP might be increased which causes aggregates to be rimed, reducing their complexity. Contrarily, the lower the present LWP, the less riming can occur and hence the more complex the shapes of aggregates can be.





Panel (d) **I** can be used to explain the decrease in particle fall velocity in combination with $Z_e$ increase: the same combination of $Z_e$ and $MDV$ is combined here with $\Delta sZ_{DR_{max}}(below - above)$. The most significant increase in $\Delta sZ_{DR_{max}}(below - above)$ coincides with the strongest increase in $Z_e$ and moderate decrease in particle fall velocity, but in the entire quadrant I, $sZ_{DR_{max}}(below - above)$ generally tends to increase. Aggregates do not produce high $Z_{DR}$, therefore the increased $\Delta sZ_{DR_{max}}(below - above)$ points to a second particle mode of smaller, anisotropic ice at the slow edge of the Doppler spec-

trum which is formed inside the turbulent layer and causes the particle population as a whole (represented by $MDV$) to fall slower. A logical explanation would be the breakup of aggregates inside the turbulent layer through ice collisional fragmentation. This hypothesis is further supported by panel (g) **I**, where LIMRAD94 $\Delta K_{DP}$ was added as color. $K_{DP}$ in quadrant **I** is mostly increasing inside the turbulent layer, which indicates a higher number of non-spherical particles, consistent with the hypothesis above. A small fraction of the $K_{DP}$ increase might also be attributed to the presence of aggregates, as these can be

responsible for up to 20 % of W-band $K_{DP}$ (Kötsche et al., 2025). In the left section of quadrant **I**, $K_{DP}$ typically exhibits minimal variation or might even slightly diminish within the turbulent layer. This is potentially due to negligible changes in particle concentration within the layer. The denser nature of rimed aggregates could reduce their likelihood of generating many splinters upon collision. Furthermore, particles rimed in the turbulent layer may also tend to become more spherical, resulting in a lower $K_{DP}$.

The temperature at TLH is shown in panel (f), for the pixels identified as containing mainly aggregates, it is very homogeneous between -12 and -15 °C. This appears to be a temperature where we see efficient aggregation inside the turbulent layer. As this is also still the temperature regime of the dendritic growth layer, it is reasonable to believe that splinters formed by ice collisional fragmentation quickly grow into dendrites and then aggregate as well. Enhanced supersaturation inside the turbulent layer might further aid new particle growth through deposition as also proposed by Lee et al. (2014) or Aikins et al. (2016).

Panel(h) **I** displays VISSS $N_{tot}$ as color. $N_{tot}$ ranges between 1 and $5\,1^{-1}$, which underlines that particle populations with many aggregates on average still contain a low number of particles because small, newly formed particles quickly aggregate as well. $N_{tot}$ is slightly higher in the left of quadrant **I**, which combined with findings from the other panels, suggests deeper warm frontal precipitation with inherently higher $N_{tot}$.

**Quadrant II**

Quadrant **II** within each sub-panel comprises data points characterized by an increase in $Z_e$ within the turbulent layer combined with a increase in $MDV$ (corresponding to higher fall velocity of particles below the turbulent layer compared to above it). This combination is the second most common during the analyzed period as evidenced in panel (i).

In panel (a) **II**, increasing of $Z_e$ in combination with a increase of $MDV$ features more data points with higher $M$, these particles are rimed and fall faster due to the density increase, however, we also see pixels where near-zero $M$ indicates unrimed

particles, especially towards the highest $Z_e$ increase. Looking at panel (b) **II**, particle populations generally feature lower $D_{32}$ values of 2 mm and smaller in some pixels to the left, but also higher $D_{32}$ especially to the right of quadrant **II**. The complexity in panel (e) **II** depicts lower values of around 1.5 in some pixels. Still, we also see many pixels with higher complexities indicating both more rimed aggregates and graupel, but also PSD mainly inheriting unrimed aggregates.



Panel (d) **II**, compared to quadrant **I**, generally indicates a smaller increase in $sZ_{DR_{max}}(below-above)$ inside the turbulent layer or even a slight decrease, especially for pixels with higher $M$. As $Z_{DR}$ is closely connected to the particle shape, this decrease may be caused by riming as it tends to render small particles more spherical. For the same reason, $K_{DP}$ in the left half of panel (g) **II** displays little change or slight decrease inside the turbulent layer for more rimed PSDs. Pixels in the right part of **II** indicate an increase of $sZ_{DR_{max}}(below-above)$ and $K_{DP}$, cross checking with the other panels, these are also pixels where $D_{32}$ is enhanced and $M$ is low, which are likely particle populations consisting of aggregates producing splinters via ice collisional fragmentation as in quadrant **I**. We may also see PSDs with coexisting aggregates and graupel particles which we described in Kötsche et al. (2025). The turbulent layer may produce pockets of SLW, causing some particles to be heavily rimed while others experience less riming (e.g. Medina and Houze, 2015). This can lead to more efficient ice collisional fragmentation through greater differences in fall velocity and density. These collisions were found to produce up to 500 fragments per collision by Grzegorczyk et al. (2023). In panel (f) **II**, it is noteworthy that for many pixels with high $M$, temperature at TLH is below -15 °C, although the overall number of datapoints is low (see panel (i)). This result differs from previous studies like Kneifel and Moisseev (2020) who found that riming is very uncommon at temperatures colder than -10 °C. The reason is likely that they did not analyze sites influenced by orographic turbulence like in this study. As we did show in Fig. 7, SLW close to Gothic Mountain Summit can be found at temperatures down to -20 °C (see Fig. 7). This implies the turbulent layer is producing SLW even at temperatures way below -10 °C and hence enables riming.

In panel (h) **II**, VISSS $N_{tot}$ interestingly shows very high values for the pixels with high $M$ and low $D_{32}$, indicating high number concentration of small graupel particles. This is consistent to the findings of Lee et al. (2014) who showed that turbulence can lead to a relatively large number of small-sized graupel particles.

**Quadrant III**

Panel (a) **III** inherits a combination of decreasing $Z_e$ and increasing $MDV$ (meaning that the particles below the turbulent layer inherit higher fall velocity compared to the ones above it). This combination appears quite infrequently (see panel (i) **III**) and seems physically atypical, as an increase in particle fall velocity usually indicates a size and/or density increase. The reduction in $Z_e$ is up to -10 dB km$^{-1}$. Quadrant **III** features particle populations with very diverse rime mass fractions (panel a) and variable $D_{32}$ between 1.5 and 2.5 mm, while pixels with below 2 mm prevail (panel (b) **III**). $sZ_{DR_{max}}$ and $K_{DP}$ both show decreasing tendencies in the turbulent layer (panel (d) and (g) **III**), which indicates that small, non-spherical particles are depleted, a signal which may be attributed to sublimation. VISSS $N_{tot}$ in panel (h) **III** agrees with that hypothesis, as the particle populations mostly contain less than 2 l$^{-1}$. As the small particles in the PSD vanish, $MDV$ will indicate faster falling particles because now the remaining larger particles dominate the spectrum. Given the very low number concentrations, spatial inhomogeneity may also be a reason for the observed behavior of $Z_e$ and $MDV$.

**Quadrant IV**

Panel (a) **IV** inherits a combination of decreasing $Z_e$ and decreasing $MDV$ (meaning particles below the turbulent layer fall slower than above it), such situations were rarely observed (panel i). Counterintuitively, we see a rather high fraction



of more rimed PSDs (higher $M$). Especially towards a stronger $Z_e$ decrease, $D_{32}$ of below $2\,\mathrm{mm}$ (panel (b) **IV**) indicates smaller particles dominating the PSD. Also, the complexity is lower than 1.6 for these particle populations (panel (e) **IV**). $sZ_{DR_{max}}$ and $K_{DP}$ both exhibit a slight decreasing or indifferent tendency, which corroborates the depletion of small particles. Nevertheless, it is plausible that both metrics were initially close to zero, as small, rimed particles with low $N_{tot}$ are unlikely to generate pronounced signals in $K_{DP}$ or $Z_{DR}$. VISSS $N_{tot}$ in panel h **III** indicates low number concentrations of below $2\,\mathrm{l}^{-1}$. Combined, these signatures points to sublimating small and slightly rimed particles, probably even graupel, without presence of larger particles. As these particles sublimate, they lose mass causing them to fall slower, without the presence of larger particles as in quadrant **III** this causes a decrease in fall velocity across the whole PSD.





## 4 Summmary and conclusions

In this study, we conducted a long-term observational study during the SAIL campaign in Gothic, Colorado. We used a unique and closely collocated instrument setup combining a slanted polarimetric W-band radar (LIMRAD94), a vertically pointing ARM Ka-band radar (KAZR), the Video In Situ Snowfall Sensor (VISSS) and a high resolution spectral lidar (HRSL) as well as other instruments to investigate the characteristics of orographic turbulent layers based on a case study (Sec. 3.1) and long-term statistical analysis (Sec. 3.2). To draw conclusions on cloud microphysical processes inside the turbulent layer, we analysed the data of LIMRAD94 and KAZR data using a novel method to assess microphysical processes by comparing radar variables above and below the turbulent layer, avoiding direct in-layer radar signal degradation from turbulence (Sec. 3.3). These results are shown in Fig. 10 and 11.

To conclude our study, we would like to highlight the following key findings:

- A turbulent layer was almost always present at the site in Gothic during precipitation events between Sep 2021 and May 2023. The temperature at TLH during these events was mostly between -5 °C and -20 °C - a favorable region for aggregation, riming, and secondary ice production (SIP). Precipitation from shallow clouds in connection with a turbulent layer was also frequently seen in this statistical analysis, which underlines the importance of orographic turbulence for precipitation formation (Fig. 9).

- The case study on Feb 21st, 2023 demonstrated turbulence-driven mixing, SLW formation, and precipitation generation even in shallow clouds forming in response to the turbulent layer (Figs. 3,4,5).

- Aggregation is a dominant process inside the turbulent layer and frequently occurs at temperatures between -12 and -15 °C (temperature at turbulent layer height). It is responsible for $Z_e$ increase of up to $20\,\mathrm{dBZ\,km^{-1}}$ and reduction of the mean particle fall velocity. This reduction is connected to the formation of small, anisotropic particles through SIP. This theory is supported by increasing $sZ_{DR_{max}}$ and increased $K_{DP}$ inside the turbulent layer in connection to the aggregation signals (Fig. 11). This indicates SIP through ice-collisional fragmentation producing new, anisotropic ice splinters that may quickly grow into dendrites, given the favorable temperature regime for depositional growth. These results are consistent with the conclusions presented in Chellini and Kneifel (2024).

- Riming frequently occurs inside the turbulent layer, $Z_e$ may increase by up to $15\,\mathrm{dBZ\,km^{-1}}$ while particle fall speeds always increase. Changes in $K_{DP}$ or $sZ_{DR_{max}}$ during riming cases with high rime mass fraction $M$ are mostly minor with a slight decreasing tendency, likely due to particles becoming more spherical. We frequently detected riming where the temperature at turbulent layer height (TLH) was below -15 °C. This implies the turbulent layer is producing SLW even at temperatures way below -10 °C and hence enables riming, contrary to the findings of Chellini and Kneifel (2024) who found riming mainly at temperatures warmer than -10 °C.

- We observe sublimation below the turbulent layer. This is characterized by a decrease in $Z_e$, $K_{DP}$ and $sZ_{DR_{max}}$ in combination with an increase or decrease in $MDV$. Often, these sublimation cases feature enhanced $M$. We speculate





that small graupel is forming in a shallow cloud layer inside the turbulent layer and sublimates in the well mixed boundary layer as presented in the case study in Sect. 3.1. Sublimation in connection to turbulence was also reported by (e.g. Lee et al., 2014; Ramelli et al., 2021).

–   Statistical analysis showed that TLH is strongly tied to the surrounding terrain and the wind direction at turbulent layer height (see Fig. 8). TLH is often collocated with the cloud base height and supercooled liquid water layers near the summit height of Gothic Mountain (Fig. 6). This implies the turbulent layer is producing SLW which was also found by many other studies in the past (e.g. Houze and Medina, 2005; Houze Jr., 2012; Medina and Houze, 2015; Ramelli et al., 2021, among others). The frequent collocation of cloud base height and TLH may be further explained by mechanical
mixing of the boundary layer due to turbulence and increased moisture convergence in the lee of Gothic Mountain.

The authors note that the measurement location has intricate terrain and unique synoptic forcing mechanisms. Therefore, the findings are particular to this specific environmental and synoptic scenario and may not be applicable to different locations or situations. Nevertheless, the presence of analogous mountainous and valley structures throughout the Rocky Mountains suggests that comparable processes are likely to occur, which may result in notable congruence with our findings. We particularly
highlight a recent campaign named S2noCliME (Pettersen et al., 2024) in the Rocky Mountains. A wide facet of instruments including multiple radar frequencies was deployed in the Park Range of Northwest Colorado during the 2024 – 2025 winter season. The data gathered there may provide further insights into the microphysical processes connected to orographic turbulence and we strongly recommend exploiting the data set with this special focus on orographic turbulence.

Our findings highlight the importance of orographic turbulence in precipitation enhancement, even in shallow clouds or
without the presence of deep precipitating systems. Precipitation from these shallow clouds may be completely caused or at least enhanced by orographic turbulence, we cannot provide conclusive evidence. This topic should be addressed in future studies. We demonstrated how radar polarimetry may be used to investigate microphysical processes inside a turbulent layer avoiding the dampening effects of turbulence on radar polarimetry.



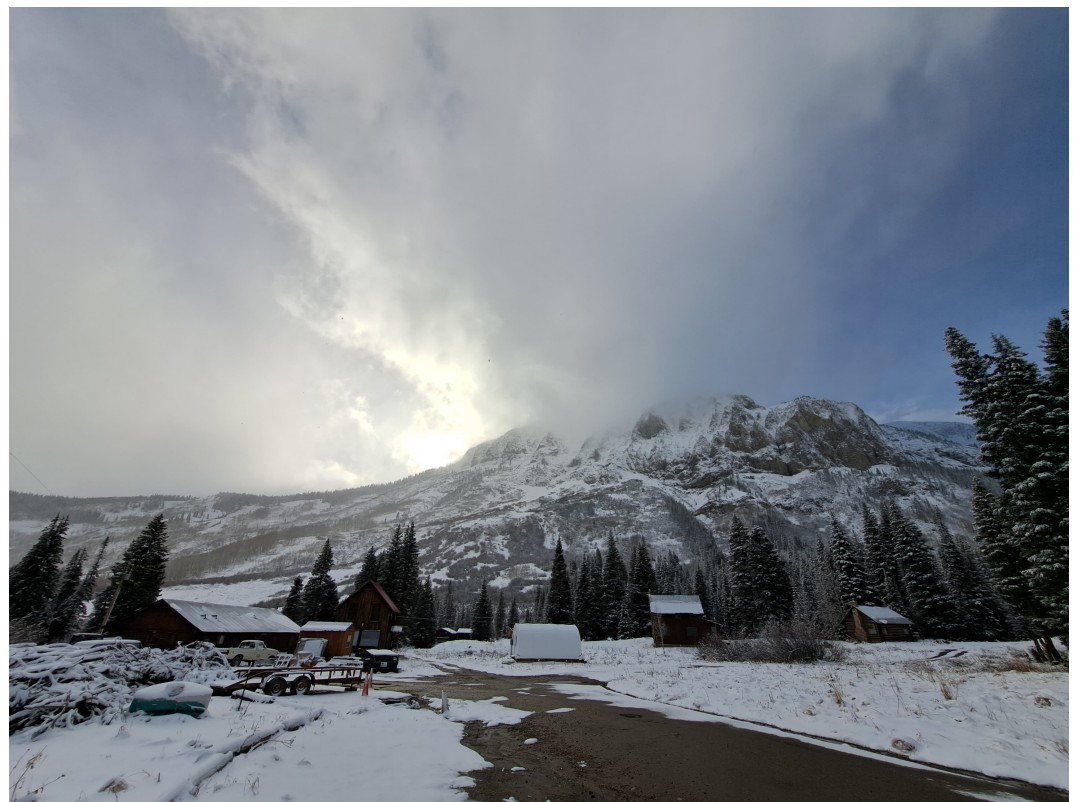

**Figure A1.** Image of a precipitating cloud behind Gothic Mountain on Oct 22, 2022 at 16:42 UTC. Courtesy: Anton Kötsche

**Appendix A: On site images of cloud formation behind Gothic Mountain**

Figure A1 shows a precipitating cloud forming behind Gothic Mountain, we observed these clouds very frequently on site during the installation of our instruments.



*Author contributions.* AK collected and processed the LIMRAD94 data from CORSIPP, analyzed and plotted the data and wrote the manuscript with contributions from all coauthors. All authors reviewed and edited the manuscript. The AI tool TeXGTP was used to correct spelling and typesetting.

*Competing interests.* The authors declare that they have no competing interests.

*Code and data availability.* SAIL data were obtained from the Atmospheric Radiation Measurement (ARM) user facility, a U.S. Department of Energy (DOE) Office of Science user facility managed by the Biological and Environmental Research Program.: LIMRAD94 (https://doi.org/10.5439/2229846, last access: December 5, 2024), VISSS (https://doi.org/10.5439/2278627, last access: December 5, 2024), the meteorological in situ 415 data of AMF2 (https://doi.org/10.5439/1786358, last access: December 5, 2024), the microwave radiometer
retrieval products (https://doi.org/10.5439/1027369, last access: 5 Dec 2024), the ARM KAZR (https://doi.org/10.5439/1498936, last access: 5 Dec 2024), the ARM high resolution spectral lidar (https://doi.org/10.5439/1462207, last access: 5 May 2025), the ARM RWP915 (https://doi.org/10.5439/1993735, last access: 5 May 2025), the ARM Parsivel (https://doi.org/10.5439/1973058, last access: 8 Aug 2025), the ARM ceilometer (https://doi.org/10.5439/1497398, last access: 10 Sept 2025)

*Acknowledgements.* This research is funded by the Deutsche Forschungsgemeinschaft (DFG, German Research Foundation) - Project num-
ber: 408008112 (Characterization of orography-influenced riming and secondary ice production and their effects on precipitation rates using radar polarimetry and Doppler spectra - CORSIPP) within the Priority Program SPP 2115 „Polarimetric Radar Observations meet Atmospheric Modelling (PROM) – Fusion of Radar Polarimetry and Numerical Atmospheric Modelling Towards an Improved Understanding of Cloud and Precipitation Processes". The paper was funded by the Open Access Publishing Fund of Leipzig University supported by the German Research Foundation within the program Open Access Publication Funding. This research was supported by the Atmospheric Radi-
ation Measurement (ARM) user facility, a U.S. Department of Energy (DOE) Office of Science user facility managed by the Biological and Environmental Research Program. This work was supported in part by the European Space Agency under the activity WInd VElocity Radar Nephoscope (WIVERN) Phase A Science and Requirements Consolidation Study, ESA Contract Number 4000144120/24/NL/IB/ab. We would like to thank Teresa Vogl for the help in processing $EDR$ data used in this manuscript. We would like to thank the Rocky Mountain Biological Laboratory staff as well as ARM technicians for logistical and on-site technical support.





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
