# Peer review of "Snow microphysical processes in orographic turbulence revealed by cloud radar and in situ snowfall camera observations"

_EGUsphere, 2025_

## Referee Comment (RC1)

**Atmospheric Chemistry and Physics - Manuscript ACP-2025-4517:**

"Snow microphysical processes in orographic turbulence revealed by cloud radar and in situ snowfall camera observations" by A. Kötsche, M. Maahn, V. Ettrichrätz and H. Kalesse-Los

**1 Summary**

This manuscript investigates the influence of a turbulent layer in the wake of the Gothic Mountain (in the Rockies) on the microphysical properties of ice crystals and snowflakes, by means of a vertically pointing K-band radar and a W-band profiler shooting at 40° elevation (in the direction of the K-band system). In addition, a high-resolution spectral lidar, a wind profiler and radiosoundings provide complementary meteorological variables.

Depending on the wind direction, the Gothic Mountain is generating a turbulent layer in its wake, and the present study analyzes its the influence on the microphysical properties of the cloud and precipitation particles, by contrasting the radar variables above and below the turbulent layer. It is concluded that (i) the turbulent layer promote aggregation (associated with strong increase in Z) and secondary ice production (SIP) via ice-ice collision (associated with decrease in the mean Doppler velocity), (ii) riming is frequent (related to the formation of supercooled liquid water droplets via turbulence), and (iii) a dry layer is frequent below the turbulent layer, sublimating the hydrometeors produced in the turbulent layer.

**2 Recommendation**

The authors take advantage of the nice instrumental set-up during the SAIL campaign, and propose an innovative approach to investigate and quantify the influence of turbulence on the properties of ice hydrometeors. They combine and cross-correlate the various data (radar moments, radar spectral variables, wind profiles from the wind profiler, liquid layer altitude from the lidar...) to pinpoint the dominant microphysical processes in this turbulent layer: aggregation, SIP and riming (not all together at the same time). The proposed approach and the obtained results are sound (although some aspects remain rather speculative), and they are relevant for the ACP readership. I do not have major concerns per se, but some suggestions of modifications or complements to improve the readability of the paper that may take some time to implement. I hence recommend to send the manuscript back to the authors for major revisions. A list of comments and questions is provided below.

**3 General comments**

1. A map showing the local topography is missing. The text refers several times (ex: p.10, l.212-213; p.13, l.261) to the shape of the Gothic Mountain, but the reader does not have

- access to this information.
- 2. Section 3.3 is important but very dense and not easy to follow. I suggest to add a schematics of the main processes and radar variables in each quadrant, to help the reader understand the various configurations and signatures involved in this interpretation.
- 3. there are various thresholds or interpretation that appear somehow arbitrary. They should be better explained and justified. See specific comments below for more details.

**4 Specific comments**

- 1. P.1, l.23-24: wind shear can be in speed, in direction or both, but not only in both (as suggested by "and" between "speed" and "direction".
- 2. P.5, l.110: what method is used to estimate  $K_{dp}$  from  $\Phi_{dp}$  and what is the final resolution of  $K_{dp}$ ?
- 3. P.5, l.127: something is wrong or a definition is missing with "level2match"...
- 4. P.5, section 2.1.2: there is no mention of possible attenuation effects at W-band (due to atmospheric gases, liquid water and even ice particles). This should be (briefly) discussed as it may influence some of the interpretations.
- 5. P.6, Fig.2: the height above ground at which the beams of KZAR and LIMRAD94 cross should be indicated (as it is constant, right?).
- 6. P.7. l.150: the LIMRAD94 sampling volume is distant of 580 m of the KAZR sampling volume used to estimate the EDR, away from the mountain (i.e. downstream from KAZR)? Could this imply that the EDR may be lower at LIMRAD gate?
- 7. P.7, Eq.2: why using a weighted mean to estimate TLH?
- 8. P.7, l.156: Define AGL here as it is the first mention (it is done later). And mention the altitude of the top of Gothic Mountain.
- 9. P.8, l.174: how is this threshold value obtained? Please justify.
- 10. P.8, l.195: this sentence is a bit confusing: you explain in details in the introduction how turbulence can affect the ice crystals, and here state that the main changes are expected below this turbulent layer. Please elaborate.
- 11. P.9, l.211: please elaborate on how the shower can influence the wind profile.
- 12. P.10. l.217-218: "favorable for depositional growth": moisture/saturation is also a key variable for deposition...
- 13. P.120, l.219: are size and complexity enough to identify graupel?
- 14. P.10, l.221 + Fig.4: are all the measured particles displayed? There seems to be a limited number of particles for such long periods (1h)... If not, how were those shown selected?
- 15. P.10, l.226: any comments about the temperature inversion at the height of Gothic Mountain top (and associated drying)?
- 16. P.10, l.228: how do you diagnose the occurrence of a MPC?

- 17. P.11, Fig.5: the green solid line figures the dewpoint temperature, not the relative humidity. And what do the green dashed lines figure (the saturation mixing ratio)?
- 18. P.11, l.238: "mostly": you have the distribution, you could be more quantitative (ex: x% of the values are below 1000m).
- 19. P.12, Fig.6: you should provide the number of data points behind the curves displayed in Fig.6. Or at least mention how many "events" occurred in the considered time period.
- 20. P.12, l.242.243: this "enhanced moisture convergence" is important but remains speculative. Any ways to strengthen this statement (high resolution NWP model runs for instance)?
- 21. P.12, l.246-247: you could compare the turbulent layer top to see if does match with LLB... Focusing on the regions of sharp negative EDR gradient above the TLH, for instance.
- 22. P.12, l.251: using the same threshold in occurrence as for the lower limit at 800m (around 0.4-0.5), you end up with an upper limit around 1800-2000 m...
- 23. P.17, l.298: particles produced by SIP would also be small with lower fall velocities. To be checked?
- 24. P.18, l.302: why no signal in Kdp? Small non-spherical ice splinters should increase Kdp, no?
- 25. P.18, l.307 + Fig.11: VISSS does not see well small particles (mentioned resolution about 60 microns), could this introduce biases?
- 26. P.18, l.319-320: "...be found in PSDs containing..": I do not understand this sentence. How do you access the PSD?
- 27. P.18, l.217, 218: a decrease, not an decrease.
- 28. P.18, l.220-224: why would the MDV decrease for such aggregates ("An increase in Ze in combination with an decrease of MDV").
- 29. P.18. l.327: visible
- 30. P.19, l.346-347: but an increased kinetic energy at the impact (so higher velocity) produces more splinters, no?
- 31. P.19, l.352: you need supersaturation with respect to liquid water for dendritic growth. Do the radiosounding indicate such supersaturation at those times/heights?
- 32. P.19, l.367-368: where is it explained how to use the complexity to identify ice crystal habits?
- 33. P.22, item 3: could the analysis of sZe strengthen the interpretation? To see if the increase in Ze is due to large (relatively) fast falling particles (i.e. the aggregates) or to many smaller ones...

---

## Author Comment (AC1)

**Author's response to:**
**RC#1**
**https://doi.org/10.5194/egusphere-2025-4517-RC1**

Anton Kötsche[1], Maximilian Maahn[1], Veronika Ettrichrätz[1], and Heike Kalesse-Los[1]

[1]Leipzig Institute for Meteorology (LIM), Leipzig University, Leipzig, Germany

**Correspondence:** anton.koetsche@uni-leipzig.de

Dear Reviewer,

Thank you for carefully reading the manuscript and pointing out several issues where the description needs to be refined for a better understanding. The requested clarifications and references to ambiguities contribute to the improvement of the manuscript.

To separate the Reviewer's comments and the author's response, we printed the comments in black and the response in blue. Excerpts of the manuscript with marked changes are pinned directly to the appropriate responses, with the indicated text location (e.g., line number) referring to the manuscript in the preprint.

Sincerely, on behalf of all authors

Anton Kötsche

**Summary of main changes of the manuscript:**

- There was a minor bug in the processing of the turbulent layer height. The EDR data was re-gridded to fit the range resolution of the ARM wind profiles, effectively halving the range resolution of EDR. This EDR was then used to process TLH, TLH was then used for all statistics presented in the manuscript. Of course, this was not necessary for most of the statistics where EDR could have been kept at full resolution. We addressed that issue and reprocessed all statistics, the overall impact however was minimal.

- Some minor changes occured in Fig. 9, there was a bug in the statistic where some of the "no turbulence" cases were falsely classified.

- A paragraph was added in L.168f. explaining the reasons for the used turbulent layer height retrieval

- As suggested by both reviewers, we created a schematic figure of main suspected processes in the turbulent layer (see Fig. 2) and added it to Sec. 3.3.

- A map of the measurement sites topography was added in the appendix of the manuscript (see Fig. B1). We also referenced the figure in Sect. 3.2.

- We removed backscattering differential phase cases from the statistics presented in Sect. 3.3 because it falsely affected overall $K_{DP}$ magnitude. In the statistic presented in Fig. 11, minor changes in $\Delta K_{DP}$ occured. Details on how backscattering differential phase were provided in Sect. 2.1.2.

- In cases with shallow precipitating clouds, when their cloud top and the turbulent layer are superimposed, primary nucleation may be especially active and we can not exclude the impact of ice nucleation on our statistic. These cases almost never end up in the statistic because no data will be present above the turbulent layer. The few cases that would have been included in the statistic were removed. Additional argumentation was added in Sect. 2.7.

[Figure]

**Figure 1.** Elevation plot of the measurement site with all relevant mountain peaks and instrument locations marked.

**Response to RC#1:**

**General comment #1: Map showing the local topography**

*A map showing the local topography is missing. The text refers several times (ex: p.10, l.212-213 ; p.13, l.261) to the shape of the Gothic Mountain, but the reader does not have 1access to this information.*

Thank you for this suggestion, we created a map of the measurement sites topography and added it in the appendix of the manuscript (see Fig. 1). We also referenced the figure in Sect. 3.2.

> **L291f.:**
>
> (see Fig. B1 for a map of the local topography)

**General comment #2: Sect 3.3 schematics of the main processes**

*Section 3.3 is important but very dense and not easy to follow. I suggest to add a schematics of the main processes and radar variables in each quadrant, to help the reader understand the various configurations and signatures involved in this interpretation. We added several references to the Figure in Sect. 3.3, see lines:*

[Figure]

**Figure 2.** Conceptual diagram of the suspected (dominant) microphysical processes inside the turbulent layer, displayed similar to the four quadrants in Fig. 11. The impact of microphysical processes on radar and VISSS variables is also displayed qualitatively: Red downward facing arrows indicate a decrease of the respective variable, upward facing green arrows an increase. Note that for the radar variables ($Z_e$, $MDV$, $sZ_{DR_{max}}$ and $K_{DP}$), we can measure the change inside the turbulent layer. VISSS variables ($N_{tot}$, $C$, $D_{32}$, $M$) are measured below the turbulent layer. Their change due to the microphysical process is suspected based on the change of particle and PSD properties. If a variable is not depicted, no clear trend can be derived. The particle images were recorded by VISSS.

Thank you for suggesting this useful addition to the manuscript, we created a schematic figure (see Fig. 2) and added it to Sec. 3.3. References to the schematic Figure were added in L397, L428

**General comment #3: Arb. thresholds**

*there are various thresholds or interpretation that appear somehow arbitrary. They should be better explained and justified. See specific comments below for more details.*

We agree that some of the chosen thresholds may seem arbitrary, we will do our best to explain our reasoning replying to your specific comments.

**Specific comments**

- *P.1, l.23-24: wind shear can be in speed, in direction or both, but not only in both (as suggested by "and " between "speed" and "direction".*

- Thank you for pointing out that logic flaw, we changed the "and" to "and/or"

- *P.5, l.110: what method is used to estimate Kdp from Φdp and what is the final resolution of Kdp?*

- $K_{DP}$ was calculated from $\Phi_{DP}$ using a convolution with low-noise Lanczos differentiators with window length of 23 gates. This method provides results that are analogue to a moving window linear regression and is implemented in the python package wradlib (Heistermann et al., 2013). The final range resolution is about 12 m. The information of KDP calculation was added in Sect. 2.1.2.

- *FP.5, l.127: something is wrong or a definition is missing with "level2match"...*

- This is the name of the data product based on and as used in Maahn et al. (2024), we chose to use the same convention.

- *P.5, section 2.1.2: there is no mention of possible attenuation effects at W-band (due to atmospheric gases, liquid water and even ice particles). This should be (briefly) discussed as it may influence some of the interpretations.*

- $K_{DP}$ magnitude is not affected by attenuation, nor is $sZ_{DR_{max}}$, however $sZ_{DR_{max}}$ can be affected by differential attenuation but this is negligible at the short measurement range we are considering. KAZR $Z_e$ is corrected for gaseous attenuation by ARM. We added a clarifying sentence concerning $Z_e$:

  | **L192.:** |
  | --- |
  | In KAZR data, $Z_e$ is already corrected for gaseous attenuation (Johnson et al., 2023). |

- *P.6, Fig.2: the height above ground at which the beams of KZAR and LIMRAD94 cross should be indicated (as it is constant, right?).*

- Thank you for this suggestion, we added it in Fig. 2 in the original manuscript (see Fig. 3).

- *P.7, l.150: the LIMRAD94 sampling volume is distant of 580 m of the KAZR sampling volume used to estimate the EDR, away from the mountain (i.e. downstream from KAZR)? Could this imply that the EDR may be lower at LIMRAD gate?*

- The center of the LIMRAD94 measurement volume at Gothic Mountain summit height is about 580 m south of the KAZR sampling volume, so the distance to Gothic Mountain summit is about the same. EDR should be identical.

- *P.7, Eq.2: why using a weighted mean to estimate TLH?*

[Figure]

**Figure 3.** Updated Fig. 2

- The approach is similar to the calculation of the mean Doppler velocity, which is a commonly used approach. In the essence, we are not necessarily interested in the height of the maximum EDR (which is what a peak detection or a simple max. EDR approach would tell us). We are more interested in the true height of the center of the turbulent layer, knowing the turbulent layer may have a vertical extent of a few hundred meters. The vertical EDR profile often includes multiple peaks, hence using a max. EDR approach can lead to less accurate results. Please see Fig. 4 as an example of a typical EDR profile behind Gothic Mountain. Using the maximum EDR would result in a TLH around 3600 m AGL, which in this case would be almost at the bottom of the turbulent layer. Using a weighted mean (red line) pretty accurately catches the true center of the turbulent layer. We added the following sentence in the manuscript:

> **L168f.:**
>
> This methodology is similar to the calculation of the mean Doppler velocity, a widely applied technique in radar meteorology. Our aim is to determine the representative height of the center of the turbulent layer, acknowledging that such layers often span several hundred meters in the vertical. Vertical $EDR$ profiles frequently exhibit multiple local maxima, therefore peak-finding algorithms can introduce substantial uncertainty. Also, the absolute maximum $EDR$ value does not necessarily coincide with the layer center. Using a mean $EDR$ weighted TLH therefore provides a more reliable estimate by accounting for the vertical distribution of $EDR$ rather than relying on isolated peaks.

- *P.7, l.156: Define AGL here as it is the first mention (it is done later). And mention the altitude of the top of Gothic Mountain.*

- Thank you, we defined it first at L171 now.

[Figure]

**Figure 4.** Examplary EDR profile behind Gothic Mountain.

- *P.8, l.174: how is this threshold value obtained? Please justify.*

- We mention this threshold also in Sect. 2.4, it is the same one that Vogl et al. (2022) used to mask turbulence, as we use the same retrieval and radar as they did this threshold is justified.

- *P.8, l.195: this sentence is a bit confusing: you explain in details in the introduction how turbulence can affect the ice crystals, and here state that the main changes are expected below this turbulent layer. Please elaborate.*

- Agreed, this sentence is confusing. Of course we meant that the main changes leading to inhomogeneities in precipitation occur inside the turbulent layer, so it is more important that the measurement volumes are closely aligned below the turbulent layer. We adapted the sentence:

> **L214f.:**
>
> Since the most significant changes in precipitation  are expected within the turbulent layer,  any resulting inhomogeneities in snow properties most likely occur below it, where LIMRAD94 and KAZR measurement volumes are closely aligned. We therefore argue that the combined dataset remains highly informative and suitable for the intended analysis.

- *P.9, l.211: please elaborate on how the shower can influence the wind profile.*

- We assume that the downdraft of the shower temporarily changed the wind direction to more north-westerly directions and also increased the wind speed in the boundary layer. We adapted the sentence:

> **L241f.**:
> ---
> Between 7 and 9  UTC a stronger shower  likely modifies the wind profile through its downdraft, slightly raising the TLH.

– *P.10. l.217-218: "favorable for depositional growth": moisture/saturation is also a key variable for deposition...*

– True, we mean that in this temperature range, the Wegener-Bergeron-Findeisen process is most effective which causes the depositional growth of ice on expense of liquid water. We clarified in the text:

> **L248f.**:
> ---
> This temperature range, favorable for depositional growth of ice particles due to the Wegener-Bergeron-Findeisen Process (WBF, (Wegener, 1911; Bergeron, 1935)), combined with vertical air motion induced by turbulence causes these clouds to almost constantly produce precipitation, as shown by the VISSS particle number spectrogram in panel (d).

– *P.120, l.219: are size and complexity enough to identify graupel?*

– Aggregates tend to be larger than the average graupel from shallow convection, but also the complexity is a solid indicator (Garrett and Yuter, 2014). We also show in the VISSS images in Fig. 4. that there is mainly graupel present.

– *P.10, l.221 + Fig.4: are all the measured particles displayed? There seems to be a limited number of particles for such long periods (1h)... If not, how were those shown selected?*

– Particle images are selected based on the sharpness of the image and their size. To blurred or to small particles are not shown. Also, there is a limit in how many particles are displayed in the box, correct. Particles in the box are plotted from big (top) to small (bottom).

– *P.10, l.226: any comments about the temperature inversion at the height of Gothic Mountain top (and associated drying)?*

– Subsidence is definitely present and the associated inversion is at the top of the cloud layer, so yes, entrainment is probably occurring. However, this is not the primary focus of the analysis.

– *P.10, l.228: how do you diagnose the occurrence of a MPC?*

– We concluded this based on the fact that we see a liquid layer in the cloud and at the same time detect ice particles at the ground which can only originate from this cloud.

– *P.11, Fig.5: the green solid line figures the dewpoint temperature, not the relative humidity. And what do the green dashed lines figure (the saturation mixing ratio)?*

– Thank you for pointing out the faulty description of the image, we fixed that. Green dashed lines are lines of equal saturation mixing ratio, correct. As this is a standard line in the Skew-T plot, we did not add a label.

– *P.11, l.238: "mostly": you have the distribution, you could be more quantitative (ex: x% of the values are below 1000m).*

– Thank you for this suggestion, we changed the sentence as following:

> **L271f.:**
>
>  In 65 % of the cases included in the statistic, the difference is less then 1000 m, showing a close collocation of LLB, CBH and TLH.

– *P.12, Fig.6: you should provide the number of data points behind the curves displayed in Fig.6. Or at least mention how many "events" occurred in the considered time period.*

– We added the following sentence:

> **L269f.:**
>
> In total, 63940 time stamps (5 min resolution) are included in the statistic.

– *P.12, l.242.243: this "enhanced moisture convergence" is important but remains speculative. Any ways to strengthen this statement (high resolution NWP model runs for instance)?*

– Unfortunately, we cannot provide more than speculations here.

– *P.12, l.246-247: you could compare the turbulent layer top to see if does match with LLB... Focusing on the regions of sharp negative EDR gradient above the TLH, for instance.*

– We appreciate the reviewer's suggestion to compare the turbulent-layer top with the liquid-layer-base (LLB) height by examining regions of sharp negative EDR gradients above the TLH. While such an analysis is in principle possible, implementing it in a robust and physically consistent way would require substantial additional work. Moreover, the LLB height plays a comparatively minor role in the context of our study, which focuses primarily on microphsical processes within and below the turbulent layer. For this reason, we believe that a detailed LLB analysis would add limited value to the main objectives of the paper.

– *P.12, l.251: using the same threshold in occurrence as for the lower limit at 800m (around 0.4-0.5), you end up with an upper limit around 1800-2000 m..*

– Agreed, we changed the lower limit from 800 to 900 m

– *P.17, l.298: particles produced by SIP would also be small with lower fall velocities. To be checked?*

– This is what we investigate in the following chapter, yes.

– *P.18, l.302: why no signal in Kdp? Small non-spherical ice splinters should increase Kdp, no?*

– Yes, but this positive change in $K_{DP}$ cannot be linked to MDV change in the histogram, thats why we do a detailed analysis in the quadrants.

- *P.18, l.307 + Fig.11: VISSS does not see well small particles (mentioned resolution about 60 microns), could this introduce biases?*

- Missing the very smaller particles might of course reduce the total number concentration of particles we can observe, which is of less interest for our study.

- *P.18, l.319-320: "...be found in PSDs containing..": I do not understand this sentence. How do you access the PSD?*

- The PSD is provided by VISSS, quantities like D32 or mean complexity are based on the measured VISSS PSD.

- *P.18, l.317, 318: a decrease, not an decrease.*

- Thank you, we fixed that everywhere in the text.

- *P.18, l.320-324: why would the MDV decrease for such aggregates ("An increase in Ze in combination with an decrease of MDV"*

- This is explained in L335f.

- *P.18. l.327: visible*

- fixed.

- *P.19, l.346-347: but an increased kinetic energy at the impact (so higher velocity) produces more splinters, no?*

- Yes, but when looking at literature comparing splinter generation between graupel-graupel and graupel-snowflake collision (e.g. Grzegorczyk et al. (2023)) we find that collision of two graupel particles produces much less splinters.

- *P.19, l.352: you need supersaturation with respect to liquid water for dendritic growth. Do the radiosounding indicate such supersaturation at those times/heights?*

- Yes they do. Please find an analysis of the max. relative humidity over ice inside TLH +/- 600 m in Fig. 5. Nearest neighbour was selected between sonde and radar data, tolerance +/- 3h. A sentence concerning supersaturation was added in Sect. 3.3. The following code was used for the calculation of RH ice

> **L391f.:**
>
> We analyzed radiosonde profiles measured during cases included in this statistic, almost all of them indicated supersaturation with respect to ice in the turbulent layer (not shown)

```
def relative_humidity_ice(RH_water, T):
    e_s_water = saturation_vapor_pressure_water(T)
    e_s_ice = saturation_vapor_pressure_ice(T)
    RH_ice = RH_water * (e_s_water / e_s_ice)
    return RH_ice
```

[Figure]

**Figure 5.** Max. relative humidity over ice inside TLH +/- 600 m where $\Delta Z_e$ >0 and $\Delta MDV$ <0

- *P.19, l.367-368: where is it explained how to use the complexity to identify ice crystal habits?*

- This can be for example found in Garrett and Yuter (2014). We added the following sentence in the VISSS description chapter:

  > **L145f.:**
  >
  > The complexity can be used to distinguish between ice particle habits as for example demonstrated in Garrett and Yuter (2014).

- *P.22, item 3: could the analysis of sZe strengthen the interpretation? To see if the increase in Ze is due to large (relatively) fast falling particles (i.e. the aggregates) or to many smaller ones...*

- This could strengthen the argumentation, agreed. However, this analysis would be rather time expensive to perform. And because of the VISSS, we can prove the presence of aggregates connected to the $Z_e$ increase. And if $Z_e$ is proportional to $D^4$ and even $D^6$ for more rimed particles, the largest particles must be driving $Z_e$.

**References**

Bergeron, T.: On the physics of clouds and precipitation, Proc. 5th Assembly UGGI, Lisbon, Portugal, 1935, pp. 156–180, https://cir.nii.ac.jp/crid/1573105975504427392, 1935.

Garrett, T. J. and Yuter, S. E.: Observed influence of riming, temperature, and turbulence on the fallspeed of solid precipitation, Geophysical Research Letters, 41, 6515–6522, https://doi.org/10.1002/2014GL061016, _eprint: https://onlinelibrary.wiley.com/doi/pdf/10.1002/2014GL061016, 2014.

Grzegorczyk, P., Yadav, S., Zanger, F., Theis, A., Mitra, S. K., Borrmann, S., and Szakáll, M.: Fragmentation of ice particles: laboratory experiments on graupel–graupel and graupel–snowflake collisions, Atmospheric Chemistry and Physics, 23, 13 505–13 521, https://acp.copernicus.org/articles/23/13505/2023/, publisher: Copernicus Publications Göttingen, Germany, 2023.

Heistermann, M., Jacobi, S., and Pfaff, T.: An open source library for processing weather radar data (wradlib), Hydrology and Earth System Sciences, 17, 863–871, https://hess.copernicus.org/articles/17/863/2013/, publisher: Copernicus Publications Göttingen, Germany, 2013.

Johnson, K., Jensen, M., and Giangrande, S.: Active Remote Sensing of CLouds (ARSCL) product using Ka-band ARM Zenith Radars (ARSCLKAZR1KOLLIAS), 2021-09-01 to 2023-06-06, ARM Mobile Facility (GUC), Gunnison, CO; AMF2 (main site for SAIL) (M1) [data set], https://doi.org/10.5439/1228768, publication Title: Atmospheric Radiation Measurement (ARM) user facility, 2023.

Maahn, M., Moisseev, D., Steinke, I., Maherndl, N., and Shupe, M. D.: Introducing the Video In Situ Snowfall Sensor (VISSS), Atmospheric Measurement Techniques, 17, 899–919, https://doi.org/10.5194/amt-17-899-2024, publisher: Copernicus GmbH, 2024.

Vogl, T., Maahn, M., Kneifel, S., Schimmel, W., Moisseev, D., and Kalesse-Los, H.: Using artificial neural networks to predict riming from Doppler cloud radar observations, Atmospheric Measurement Techniques, 15, 365–381, https://doi.org/10.5194/amt-15-365-2022, publisher: Copernicus GmbH, 2022.

Wegener, A.: Thermodynamik der atmosphäre, J. A. Barth, https://books.google.de/books?id=BMMxAAAAMAAJ, 1911.

---

## Author Comment (AC2)

**Author's response to:**
**RC#2**
**https://doi.org/10.5194/egusphere-2025-4517-RC2**

Anton Kötsche[1], Maximilian Maahn[1], Veronika Ettrichrätz[1], and Heike Kalesse-Los[1]

[1]Leipzig Institute for Meteorology (LIM), Leipzig University, Leipzig, Germany

**Correspondence:** anton.koetsche@uni-leipzig.de

Dear Reviewer,

Thank you for carefully reading the manuscript and pointing out several issues where the description needs to be refined for a better understanding. The requested clarifications and references to ambiguities contribute to the improvement of the manuscript.

To separate the Reviewer's comments and the author's response, we printed the comments in black and the response in blue. Excerpts of the manuscript with marked changes are pinned directly to the appropriate responses, with the indicated text location (e.g., line number) referring to the manuscript in the preprint.

Sincerely, on behalf of all authors

Anton Kötsche

**Summary of main changes of the manuscript:**

– There was a minor bug in the processing of the turbulent layer height. The EDR data was re-gridded to fit the range resolution of the ARM wind profiles, effectively halving the range resolution of EDR. This EDR was then used to process TLH, TLH was then used for all statistics presented in the manuscript. Of course, this was not necessary for most of the statistics where EDR could have been kept at full resolution. We addressed that issue and reprocessed all statistics, the overall impact however was minimal.

– Some minor changes occured in Fig. 9, there was a bug in the statistic where some of the "no turbulence" cases were falsely classified.

– A paragraph was added in L.168f. explaining the reasons for the used turbulent layer height retrieval

– As suggested by both reviewers, we created a schematic figure of main suspected processes in the turbulent layer (see Fig. 6) and added it to Sec. 3.3.

– A map of the measurement sites topography was added in the appendix of the manuscript (see Fig. B1). We also referenced the figure in Sect. 3.2.

– We removed backscattering differential phase cases from the statistics presented in Sect. 3.3 because it falsely affected overall $K_{DP}$ magnitude. In the statistic presented in Fig. 11, minor changes in $\Delta K_{DP}$ occured. Details on how backscattering differential phase were provided in Sect. 2.1.2.

– In cases with shallow precipitating clouds, when their cloud top and the turbulent layer are superimposed, primary nucleation may be especially active and we can not exclude the impact of ice nucleation on our statistic. These cases almost never end up in the statistic because no data will be present above the turbulent layer. The few cases that would have been included in the statistic were removed. Additional argumentation was added in Sect. 2.7.

**Response to RC#2:**

*I am very skeptical on the interpretation of shallow precipitation cases where the turbulent layer almost overlays with cloud top. Since you want to get snow microphysical signatures, your basic assumption is that the turbulence affects snow growth and ice nucleation should play a minor role. However, the turbulent layer is so close to the cloud top where ice nucleation is actively taking place that you cannot exclude the impact of ice nucleation. In this regard, you are not comparing snow before and after entering the turbulent layer. You may check Chellini, G. and Kneifel 2024, and see how you can disentangle the impact of turbulence. I would suggest remove shallow precipitation cases. You may discuss how turbulence affects snow FORMATION in a separate study, but not here.*

Thank you for pointing out this very important issue with our analysis. We are aware of the general impact of INP and nucleation in the turbulent layer, however, we assume this effect to be comparably small for two reasons. The first is that the shallow precipitation cases almost never end up in the statistic presented in the main Figure 11 (28 out of 1925 time stamps used in the analysis). The reason is that when comparing data above and below the turbulent layer, no data will be present above the turbulent layer if the turbulent layer is near the cloud top. For identifying shallow precipitation cases we used the same criteria as in Figure 9 of the manuscript. We removed the shallow cases from the statistics. The second reason is related to the number of INPs present at the site. A very recent study (preprint) on the INP concentration during SAIL revealed that INP number concentrations are indeed very low in the temperature region we are interested in. Please see Figure 5 in Zhou et al. (2025), we also took a screenshot of this figure and included it here for you to look at (see Fig. 1). At temperatures around -15 °C or warmer, the INP concentration is $10^{-2}\,\mathrm{L}^{-1}$ or lower. We included this argumentation in the paper draft so improve out argumentation.

> **L218f.:**
>
> We are furthermore aware that ice nucleation through ice nucleating particles (INP) may constantly occur inside the turbulent layer in parallel to SIP and thereby adulterate the analysis. However, a recent study of Zhou et al. (2025) on the INP concentrations during the SAIL campaign found the INP concentrations during the cold season to be $10^{-2}\,\mathrm{L}^{-1}$ or less at temperatures of -15 °C or warmer, which is the primary temperature region we find the turbulent layer in during our analysis. Therefore, the impact of primary ice nucleation on the total number concentration can assumed to be small. In case of precipitation from a shallow cloud layer, where the turbulent layer is close to the cloud top, the effect of primary ice nucleation may be more significant. However, these cases almost never end up in the statistic because no data will be present above the turbulent layer. The few cases that would have been included in the statistic were removed. These cases were identified using KAZR $Z_e$, the criterion for precipitation from a shallow cloud layer was fulfilled if KAZR $Z_e$ between 500 m and 1.5 km above Gothic Mountain summit height did not exceed -10 dBZ.

[Figure]

**Figure 5.** (a) INP temperature spectra and (b) IN active surface site density (n$_s$) categorized by sampling date as cold seasons (December–March) and other seasons (April–November). n$_s$ was calculated based on the surface area of particles larger than 500 nm.

**Figure 1.** Screenshot of Figure 5 from Zhou et al. (2025)

**Comments**

- *L108 How did you compute KDP from PHIDP? At W-band, you may expect non-Rayleigh scattering. How did you deal with the contamination of differential backscatter phase shift?*

- Thank you for pointing out this important point, first of all, $K_{DP}$ was calculated from $\Phi_{DP}$ using a convolution with low-noise Lanczos differentiators with window length of 23 gates. This method provides results that are analogue to a moving window linear regression and is implemented in the python package wradlib (Heistermann et al., 2013). Backscattering differential phase $\delta$ is an important issue with our analysis. We usually see $\delta$ associated with larger graupel, occurring comparably seldom in our dataset. Due to noise in PhiDP and the comparably small number of range gates (i.e. smoothing) we applied to calculate $K_{DP}$, we may see slightly negative $K_{DP}$, especially along cloud edges, which is mostly related to non-uniform beam filling rather than $\delta$. Due to the temporal averaging we apply to the data, we assumed $\delta$ to be negligible at first, but based on your comment we had a closer look and detected the occurrence of $\delta$ using $K_{DP}$ and $Z_e$ thresholds. We defined $\delta$ to be present where $K_{DP}$ is below -0.5 $°km^{-1}$, $Z_e$ is higher than 0 dBZ and SNR is above 10 dB. We then removed all profiles where more than 10 range gates fulfilled the backscatter criterion. This means we are not just masking negative $K_{DP}$, we also remove positive $K_{DP}$ that may result of bumps in PhiDP

caused by non-Rayleigh effects. The reprocessed statistic is shown in Fig. 2. As expected, a few cases with graupel were removed, the overall change is rather small. Mostly, there is a slight increase in $K_{DP}$ in quadrant II, presumably because negative $K_{DP}$ causes by back scatter was removed. Please not that in that version of the figure, the shallow precipitation cases were also removed already. We added a section on $\delta$ in Sect 2.1.2:
* * *
**L108f.**:

The differential phase shift $\Phi_{DP}$ (°) consists of a backscatter and a propagational part, where the propagational part is called specific differential phase $K_{DP}$ (° km$^{-1}$). We calculated $K_{DP}$ from $\Phi_{DP}$ using a convolution with low-noise Lanczos differentiators with a window length of 23 gates. This method is implemented in the python package wradlib (Heistermann et al., 2013). Backscatter differential phase ($\delta$) is caused by hydrometeors large enough relative to the radar wavelength such that the scattering is in the non-Rayleigh regime (Trömel et al., 2013). $\delta$ was found to be negligible for frozen hydrometeors in the S, C, and X-bands (Balakrishnan and Zrnic, 1990; Ryzhkov et al., 2011; Trömel et al., 2013). In the data of our W-band radar however, we find a contribution of $\delta$, predominantly during cases with large graupel. Because removing the $\delta$ contribution from $\Phi_{DP}$ is not easy to perform, we chose to omit radar data where $\delta$ is occurring. $\delta$ causes so called bumps, sudden increases in $\Phi_{DP}$ followed by a decrease. This results in negative $K_{DP}$ values, which are therefore a good indicator for $\delta$. We defined $\delta$ to be present if $K_{DP}$ in a radar range gate is below -0.5 $° km^{-1}$, $Z_e$ is higher than 0 dBZ and SNR is above 10 dB. We then removed all profiles where more than 10 range gates fulfilled the $\delta$ criterion.
* * *
– *L117. Check the definition of ZDR. It is undoubtedly affected by particle concentration. I would simply remove this sentence.*

– We agree that the sentence might be misleading. $Z_{DR}$ is defined as follows:

$$Z_{DR} = 10 \cdot \log_{10} \left( \frac{Ze_H}{Ze_V} \right) \tag{1}$$

where both $Z_H$ and $Z_V$ are proportional to the number of particles, correct. But when number concentration is increased, both $Z_H$ and $Z_V$ increase by the same factor, so their ratio and hence $Z_{DR}$ stays the same. $Z_{DR}$ is therefore unaffected by the particle concentration. But of course, a certain number of particles must be present to produce $Z_e$ in the first place. To clarify, we replaced the sentence by a more precise wording:
* * *
**L128f.**:

Unlike $K_{DP}$, $Z_{DR}$ is  immune to the total concentration factor.
* * *
– *L121. Not really true. Firstly, overall ZDR is not affected by spectral broadening. Then, speaking of the spectral analysis, spectral broadening may lead to smaller maximum spectral ZDR, but is not necessarily lowering all spectral ZDR.Regarding the impact of turbulence on ZDR, I believe you are referring to more scattered canting angles in enhanced turbulence. See literature below,*

[Figure]

**Figure 2.** Updated Figure 11, backscattering differential phase cases and shallow precipitation cases removed.

– You are correct, ZDR is not affected by spectral broadening, and by more random orentation of particles we mean the width of the canting angle distribution as stated above when describing $K_{DP}$. We adapted the sentence to be more precise:

> **L132f.:**
>
> Similar to $K_{DP}$, overall $Z_{DR}$ values are reduced by turbulence through  an increased width of the canting angle distribution. $sZ_{DR_{max}}$ is furthermore reduced by broadening of the Doppler spectrum caused by turbulence.

– *L134 What is VAP?*

- This means value added product, we replaced VAP with "value added product" in the text.

- *L136 How did you merge sonding and MWR data?*

- This VAP is produced by ARM and we do not take part in the processing, however we did state that MWR temperature data is included in the retrieval. Actually, only precipitable water vapor from the MWR data is included.

> **L150f.:**
>
>  For the processing, ARM uses sounding data collected on site through two radiosonde launches per day (11 and 23 UTC) They then transform the data into continuous daily files with 1-minute time resolution and combined with ARM 3-channel microwave radiometer  precipitable water vapor data.

- *L146 In ARCTRIS framework, turbulence is estimated with O'Connor et al., 2010. What is the difference between the two approaches? Did you get consistent results from Vogl et al., 2024?*

- The two methods retrieve turbulence in different ways: O'Connor et al. (2010) estimate EDR from the short-time variance of successive mean Doppler-velocity measurements (a direct variance method), whereas Vogl et al. (2024) derive EDR from the -5/3 inertial-range slope of the mean-Doppler-velocity power spectrum over multi-minute windows (a spectral-slope method). Thus, O'Connor uses small-scale, high-frequency fluctuations, while Vogl relies on identifying an inertial subrange in the MDV spectrum. As we did not apply the retrieval of O'Connor et al. (2010) and therefore cannot comment on how consistent both results are.

- *L207 Time-height plot of EDR should be given, so that the reviewer is convinced that the EDR retrieval and turbulence layer classification are reasonable.*

- Please see Fig. 3 (b).

- *L222 I do not see evidence of sublimation from Z. A quantitative comparison is needed.*

- Please see Fig. 4 for a quantitative comparison of Ze evolution during the case study.

- *L231 It is common to see velocity oscillations at cloud tops. Why did you attribute the formation of the turbulence layer to the mountain effect?*

- Thank you for this important question, and you are right, it is of course not trivial to directly attribute the impact of Gothic Mountain on the turbulent layer. We can only assume a link to orography because of circumstantial evidence:

  1. Immediate upstream proximity of Gothic Mountain, less than 2km between summit and measurement volume.

  2. Collocation between turbulent layer and summit height.

[Figure]

**Figure 3.** Casestudy panel for 21.02.2023 with EDR plotted in panel (b)

3. Wind speed increase above Gothic mountain summit height (may also be caused because of the position of the inversion, however the inversion may have been strengthened by the turbulent mixing of the atmosphere below it. This is hard to disentangle).

4. Statistical evidence of the collocation between Gothic Mountain summit height and turbulent layer height depending on the wind direction.

– *L234. This is not a good case showing the impact of turbulence. Ideally, you may show that the turbulent layer is located in the ice growth path. However, cloud top is in the turbulent layer in this case.*

[Figure]

**Figure 4.** KAZR Ze difference between Gothic Mountain summit height and 2nd range gate above ground.

– *L238. Again, there is a critical inference issue for shallow precipitation cases. The turbulent layer almost overlays the cloud top, and ice nucleation processes are actively taking place in the turbulent layer. Then, when you are inferring ice growth processes by comparing the observations above and below the turbulent layer, you cannot rule out the role of ice nucleation and rapid deposition in local updrafts. The nice part of Chellini and Kneifel (2024) is that the turbulent layer is well below the cloud top, and the impact of ice nucleation is minimized.*

– In our opinion, especially this shallow precipitation case reveals the importance of the turbulent layer. There is no seeding particles from above, which means all the microphysics occur inside the shallow cloud layer influenced by turbulence. Temperatures inside the clouds are not cooler than -12 °C, which translates to max. 3e-3 INP per Liter (3 per $\mathrm{m}^{-3}$) looking at Fig. 1 blue dots. We now use this number as max. possible particle number through INP activation, see Fig 5. We see that Ntot during the case study even during times with weak precipitation is still 1 - 3 orders of magnitude larger than what would be possible through INP activation alone. This shows SIP is likely responsible for the remaining majority of detected particles.

– *L295 Because of the concern given above, I am afraid that interpretating statistics in Fig. 10 is not well supported.*

– As mentioned above, shallow precipitation cases almost never end up in the statistic and the remaining ones were removed. The statistic remained almost unaffected by this.

– *Fig 11. Some obvious issues. Full physical meaning of the abbreviations should be given. (a) (c). High riming should fall in high LWP regions. However, the inverse is presented. (d) (g). Hight KDP and ZDRmax can simply a result of depositional growth in the turbulent layer, instead of secondary ice. A conceptual diagram should be given.*

[Figure]

**Figure 5.** VISSS Ntot $[\mathrm{m}^{-3}]$ and max. Ntot through INP activation for case study on Feb. 21 2023

– (a) Due to limited space in the Fig, not all variables can be depicted with full name, we added the full name on the colorbar if possible. (c) The signal got a bit clearer now, however high LWP is not necessary present during riming at this site. Especially if the liquid water is produced by the turbulent layer, there is no large LWP signal but still we see increased riming. (g) yes, we agree that KDP and sZDRmax are increased by depositional growth, this is one of our arguments. However, we argue that the majority of pristine crystals originates from ice splinters produced through SIP which serve as ice embryos. Given the low INP number, we assume SIP to be the driving factor here. One graupel-snowflake collision produces up to 500 fragments per collision Grzegorczyk et al. (2023). One of these collisions per cubic meter would provide more embryos for depositional growth than $166\,m^3$ of air containing the max. number of INP found by Zhou et al. (2025).

[Figure]

**Figure 6.** Conceptual diagram of the suspected (dominant) microphysical processes inside the turbulent layer, displayed similar to the four quadrants in Fig. 11. The impact of microphysical processes on radar and VISSS variables is also displayed qualitatively: Red downward facing arrows indicate a decrease of the respective variable, upward facing green arrows an increase. Note that for the radar variables ($Z_e$, $MDV$, $sZ_{DR_{max}}$ and $K_{DP}$), we can measure the change inside the turbulent layer. VISSS variables ($N_{tot}$, $C$, $D_{32}$, $M$) are measured below the turbulent layer. Their change due to the microphysical process is suspected based on the change of particle and PSD properties. If a variable is not depicted, no clear trend can be derived. The particle images were recorded by VISSS.

**References**

Balakrishnan, N. and Zrnic, D. S.: Use of polarization to characterize precipitation and discriminate large hail, Journal of the Atmospheric Sciences, 47, 1525–1540, http://ams.allenpress.com/perlserv/?request=get-abstract&doi=10.1175%2F1520-0469(1990)047%3C1525:UOPTCP%3E2.0.CO%3B2, number: 13 Publisher: American Meteorological Society, 1990.

Grzegorczyk, P., Yadav, S., Zanger, F., Theis, A., Mitra, S. K., Borrmann, S., and Szakáll, M.: Fragmentation of ice particles: laboratory experiments on graupel–graupel and graupel–snowflake collisions, Atmospheric Chemistry and Physics, 23, 13 505–13 521, https://acp.copernicus.org/articles/23/13505/2023/, publisher: Copernicus Publications Göttingen, Germany, 2023.

Heistermann, M., Jacobi, S., and Pfaff, T.: An open source library for processing weather radar data (wradlib), Hydrology and Earth System Sciences, 17, 863–871, https://hess.copernicus.org/articles/17/863/2013/, publisher: Copernicus Publications Göttingen, Germany, 2013.

Ryzhkov, A., Pinsky, M., Pokrovsky, A., and Khain, A.: Polarimetric Radar Observation Operator for a Cloud Model with Spectral Microphysics, https://doi.org/10.1175/2010JAMC2363.1, section: Journal of Applied Meteorology and Climatology, 2011.

Trömel, S., Kumjian, M. R., Ryzhkov, A. V., Simmer, C., and Diederich, M.: Backscatter Differential Phase—Estimation and Variability, https://doi.org/10.1175/JAMC-D-13-0124.1, section: Journal of Applied Meteorology and Climatology, 2013.

Zhou, R., Perkins, R., Juergensen, D., Barry, K., Ayars, K., Dutton, O., DeMott, P., and Kreidenweis, S.: Seasonal variability, sources, and parameterization of ice-nucleating particles in the Rocky Mountain region, EGUsphere, pp. 1–48, https://doi.org/10.5194/egusphere-2025-4306, publisher: Copernicus GmbH, 2025.